# Physical processes and biological productivity in the upwelling regions of the tropical Atlantic

Peter Brandt[1,2], Gaël Alory[3], Founi Mesmin Awo[4], Marcus Dengler[1], Sandrine Djakouré[5], Rodrigue Anicet Imbol Koungue[1], Julien Jouanno[3], Mareike Körner[1], Marisa Roch[1], Mathieu Rouault[4,†]

[1]GEOMAR Helmholtz Centre for Ocean Research Kiel, Kiel, Germany
[2]Faculty of Mathematics and Natural Sciences, Kiel University, Kiel, Germany
[3]LEGOS, CNES/CNRS/IRD/UPS, Toulouse, France
[4]Nansen-Tutu Centre for Marine Environmental Research, Department of Oceanography, University of Cape Town, Cape Town, South Africa
[5]LASMES, UFR SSMT, Felix Houphouët-Boigny University, Abidjan, Côte d'Ivoire
[†]deceased

*Correspondence to*: Peter Brandt (pbrandt@geomar.de)

## Abstract

In this paper, we review observational and modelling results on the upwelling in the tropical Atlantic between 10°N and 20°S. We focus on the physical processes that drive the seasonal variability of surface cooling and upward nutrient flux required to explain the seasonality of biological productivity. We separately consider the equatorial upwelling system, the coastal upwelling system of the Gulf of Guinea and the tropical Angolan upwelling system. All three tropical Atlantic upwelling systems have in common a strong seasonal cycle with peak biological productivity during boreal summer. However, the physical processes driving the upwelling vary between the three systems. For the equatorial regime, we discuss the wind forcing of upwelling velocity and turbulent mixing as well as the underlying dynamics responsible for thermocline movements and current structure. The coastal upwelling system in the Gulf of Guinea is located along its northern boundary and is driven by both local and remote forcing. Particular emphasis is placed on the Guinea Current, its separation from the coast and the shape of the coastline. For the tropical Angolan upwelling, we show that this system is not driven by local winds, but instead results from the combined effect of coastally trapped waves, surface heat and freshwater fluxes, and turbulent mixing. Finally, we review recent changes in the upwelling systems associated with climate variability and global warming and address possible responses of upwelling systems in future scenarios.

## Short summary

Tropical upwelling systems are among the most productive ecosystems globally. The tropical Atlantic upwelling undergoes a strong seasonal cycle that is forced by the wind. Local wind-driven upwelling and remote effects particularly via the propagation of equatorial and coastal trapped waves lead to an up- and downward movement of the nitracline. Turbulent mixing results in upward supply of nutrients. Here, we review the different physical processes responsible for biological productivity.

## 1 Introduction

The tropical oceans are important to the Earth system for several reasons. The ocean receives a large part of shortwave radiation from the sun arriving at the Earth's surface that must be redistributed horizontally and vertically. Similar important exchanges of carbon dioxide, oxygen and other trace gases occur at the interface between tropical ocean and overlying atmosphere. Tropical marine ecosystems are among the most productive ones, with high relevance for global fisheries (Longhurst, 1993). They are associated with a substantial carbon

flux from the near-surface into the deep ocean (Kiko et al., 2017). At the same time, the tropical oceans are affected by modes of natural climate variability that have reverberations around the globe, e.g., including the Pacific El Niño or the Atlantic Niño. Climate warming and change are thought to profoundly affect the tropical oceans. On the one hand, they impact natural climate variability via an intensification of climate extremes or changes in natural variability (Cai et al., 2018; Crespo et al., 2022; Prigent et al., 2020b; Yang et al., 2022). On the other hand, they are thought to change the wind forcing of tropical oceans (Wang et al., 2015; Bakun, 1990) or enhance stratification thereby impacting subduction, upwelling, and air-sea gas exchange with consequences for acidification, deoxygenation (Oschlies et al., 2018) and marine ecosystems.

The zonal extent of the tropical Atlantic is similar to that of the Indian Ocean and about three times smaller than that of the Pacific Ocean. The difference in size between the Pacific and Atlantic oceans seems to be the main reason for the dominance of interannual climate variability in the tropical Pacific, while the tropical Atlantic has largest variability on seasonal time scales (Keenlyside and Latif, 2007; Burls et al., 2011). Moreover, the annual and semiannual cycles of primary production are strongly enhanced in the tropical Atlantic compared to the tropical Pacific (Mao et al., 2020). A geographical peculiarity of the tropical Atlantic Ocean is the existence of the Gulf of Guinea which, in addition to its eastern boundary, boarders to the African continent in the north, approximately along 5°N from 10°W to 10°E. There are several major rivers flowing into the tropical Atlantic including the Amazon, Congo and Niger rivers (Fig. 1).

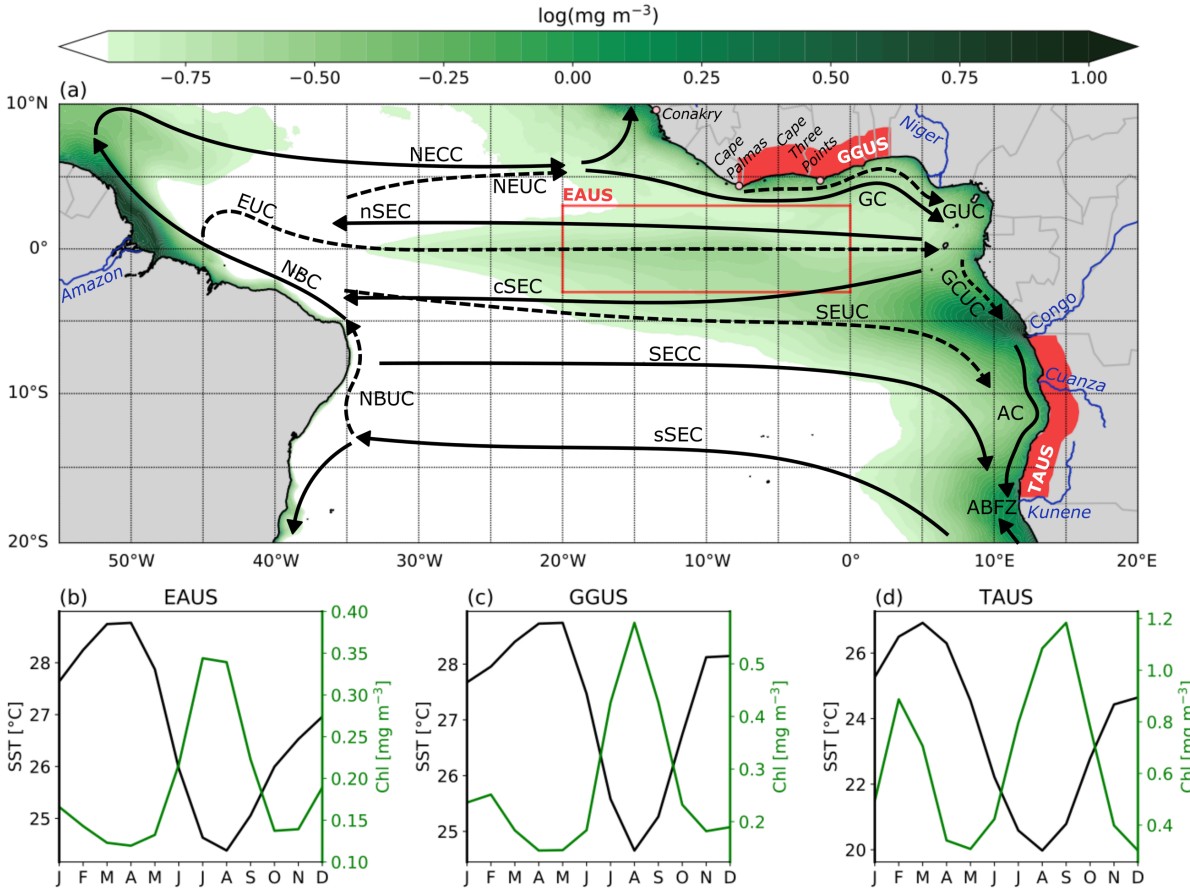

**Fig. 1** (a) Mean chlorophyll concentration in the tropical Atlantic with circulation schematic superimposed. Surface (solid arrows) and thermocline (dashed arrows) current branches shown are the North Equatorial Countercurrent (NECC), the North Equatorial Undercurrent (NEUC), the Guinea Undercurrent (GUC), the Guinea Current (GC), the North Brazil Undercurrent (NBUC), the North Brazil Current (NBC), the Equatorial Undercurrent (EUC), the northern, central and southern branches of the South Equatorial Current (nSEC, cSEC, and sSEC), the South Equatorial Undercurrent (SEUC), the South Equatorial Countercurrent (SECC), the Gabon-

Congo Undercurrent (GCUC), and the Angola Current (AC). Also marked is the Angola-Benguela Frontal Zone (ABFZ) at about 17°S and the rivers Amazon, Niger, Congo, Cuanza, and Kunene. The red box marks the equatorial Atlantic upwelling system (EAUS, 20°W-0°, 3°N-3°S). Red patches mark the coastal extent of the Gulf of Guinea upwelling system (GGUS, 8°W-3°E, 1°-width coastal band) and the tropical Angolan upwelling system (TAUS, 6°-17°S, 1°-width coastal band). The mean seasonal cycle of SST and Chlorophyll is shown for EAUS (b), GGUS (c), and TAUS (d). SST data are from the Microwave OI SST product and Chlorophyll data are from Copernicus-GlobColour both averaged for 1998-2020.

Based on satellite data, Longhurst (1993) provided a first systematic overview of the different open ocean and coastal upwelling systems in the tropical Atlantic. Today, mean satellite chlorophyll concentration (Fig. 1) reveals enhanced productivity in the different coastal upwelling regions of the tropical Atlantic such as in the Gulf of Guinea upwelling system (GGUS, here defined as 8°W-3°E, 1°-width coastal band) and in the tropical Angolan upwelling system (TAUS, here defined as 6°-17°S, 1°-width coastal band). South of the TAUS, the permanent northern Benguela upwelling system is located with the Kunene upwelling cell at about 17°S forming its northern boundary (Siegfried et al., 2019). The equatorial Atlantic upwelling system (EAUS, here defined as 20°W-0°, 3°N-3°S) is an open ocean upwelling region characterized by albeit enhanced but in comparison to the coastal upwelling systems relatively weak chlorophyll concentration (note the logarithmic scale for the chlorophyll concentration in Fig. 1) (Grodsky et al., 2008). Nevertheless, the EAUS is still of major importance for the overall biological productivity in the tropical Atlantic due to its large oceanic extent. Besides the tropical upwelling systems, enhanced chlorophyll concentration is found in the regions of the Amazon and the Congo river mouths. In the region of the Niger River mouth no comparable signal of enhanced chlorophyll is found likely due to much-reduced discharge of the Niger compared to the Amazon or Congo rivers (Fig. 1).

The tropical Atlantic upwelling is an element of the shallow overturning circulation, the subtropical cells (STCs), and is connected to the subduction in the eastern subtropics via equatorward thermocline flow and poleward Ekman transport in the surface layer (Schott et al., 2004; Fu et al., 2022; Tuchen et al., 2020). At the equator, the Equatorial Undercurrent (EUC) transports thermocline waters eastward, toward the EAUS. Due to the presence of the Atlantic meridional overturning circulation, these waters are almost exclusively of southern hemisphere origin (Schott et al., 1998; Johns et al., 2014; Tuchen et al., 2022a). Part of the waters recirculates into the westward current branches of the South Equatorial Current, the northern and the central South Equatorial Current, or contributes to supply the southward flow along the eastern boundary within the Gabon-Congo Undercurrent and the Angola Current (Kolodziejczyk et al., 2014; Kopte et al., 2017). The GGUS is supplied by the Guinea Current and the Guinea Undercurrent. While the waters of the Guinea Current mostly originate in the North Equatorial Countercurrent, a similar connection between the North Equatorial Undercurrent and the Guinea Undercurrent is less obvious (Bourlès et al., 2002; Djakouré et al., 2017; Herbert et al., 2016). Due to the much smaller width of the Atlantic compared to the Pacific, the thermocline waters arriving from the western boundary in the eastern basin upwelling systems are less enhanced in nitrate and less reduced in oxygen in the Atlantic compared to the Pacific (Brandt et al., 2015; Chai et al., 1996; Radenac et al., 2020).

The tropical Atlantic and its upwelling systems undergo a strong seasonal cycle (Fig. 1b-d). Main drivers are the seasonally varying winds associated with the meridional migration of the Intertropical Convergence Zone (ITCZ) (Fig. 2). During boreal summer, the ITCZ migrates northward resulting in strongly enhanced upwelling-favouring easterly winds along the equator and the establishment of the Atlantic cold tongue (ACT) centred around 10°W (Fig. 2c). The equatorial Atlantic is warmest in March/April (Fig. 2b) corresponding to a seasonal cycle with a fast cooling during the onset phase of the ACT and slower warming after it has reached its maximum spatial extent (Caniaux et al., 2011; Brandt et al., 2011). At the eastern boundary between equator and 15°S, lowest sea surface temperatures (SST) near the coast are found

between July and September. At the northern boundary of the Gulf of Guinea, winds strengthen as well during boreal summer in accordance with northward migration of the ITCZ and the development of the West African Monsoon resulting in upwelling-favourable westerlies along the Ghanaian coast. Lowest SST near the coast is found in July-August (Fig. 2c).

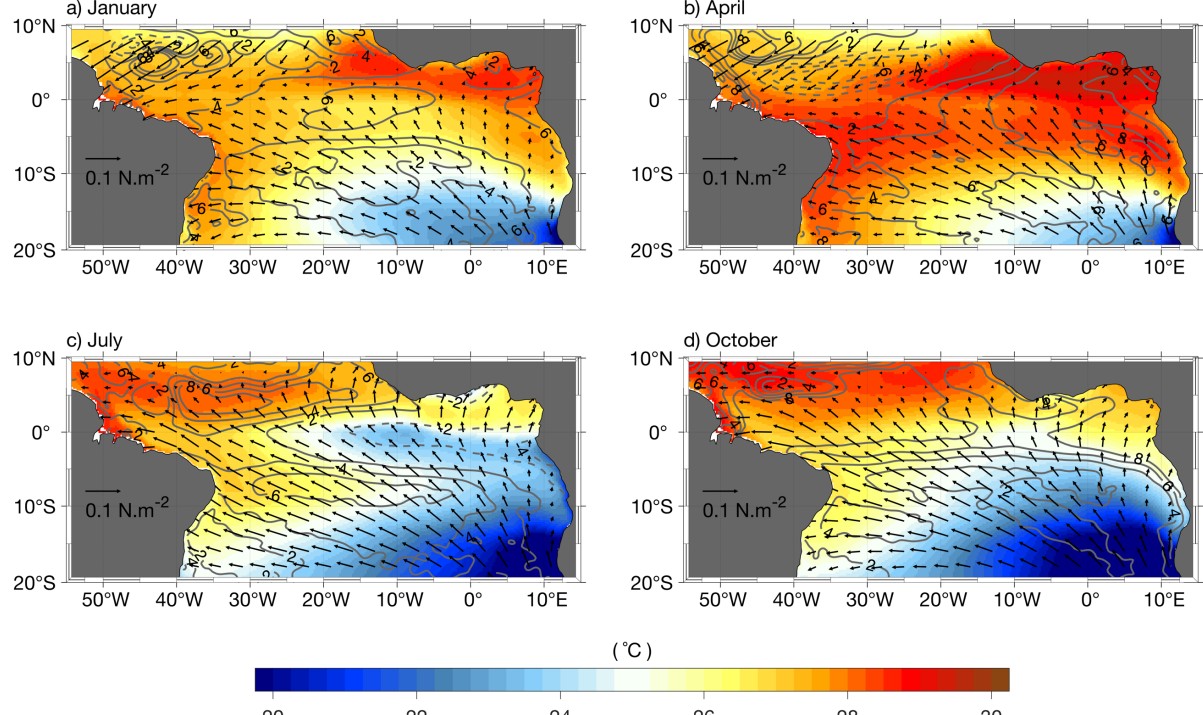

**Fig. 2** Monthly mean sea surface temperature (SST, colour shading), sea level anomaly (contour lines, unit is cm), and wind stress (arrows) during (a) January, (b) April, (c) July and (d) October. SST data are from OI-SST (https://www.esrl.noaa.gov/psd/data/gridded/), surface wind stress from ERA5 (https://cds.climate.copernicus.eu/) and sea level anomalies from the European Union Copernicus Marine Service Information (http://marine.copernicus.eu/). The data are averaged between 1982-2021.

In the tropical Atlantic, the thermocline depth can often be associated with the depth of the nitracline. An upward movement of the thermocline thus marks upward vertical advection of nitrate fuelling biological productivity (Radenac et al., 2020). Besides local wind forcing, the propagation of equatorial and coastally trapped waves (CTWs) along the equatorial and coastal waveguides, respectively, contributes to the vertical movement of the thermocline/nitracline. Such wave propagation can result in dynamic upwelling far away from the wave generation sites (Illig et al., 2018b; Illig et al., 2018a; Bachèlery et al., 2020; Hormann and Brandt, 2009). The Hovmöller diagrams of SST and winds as well as chlorophyll concentration and sea surface height show the seasonal development along the equatorial and coastal waveguides in the northern (Fig. 3) and southern (Fig. 4) hemispheres, respectively. Primary cooling in the EAUS can be identified following the enhancement of upwelling-favouring easterly winds along almost the whole equator in May-June (Weingartner and Weisberg, 1991). A secondary cooling occurs in November-December (Jouanno et al., 2011a; Okumura and Xie, 2006). In the GGUS, where SST reaches minimum values in August (Fig. 3a), upwelling-favouring westerly winds contribute to local cooling (Djakouré et al., 2017). Contrary, the southerly winds in the TAUS are particularly weak during phases of coldest sea surface (Fig. 4a) (Körner et al., 2022; Ostrowski et al., 2009). Biological productivity (or chlorophyll concentration) is generally enhanced during periods of depressed sea surface height or correspondingly during periods of elevated thermocline (Fig. 4b).

In this review, we focus on the upper-ocean seasonal cycle in the three eastern-basin upwelling systems of the tropical Atlantic between about 10°N and 20°S, the physical forcing driving the

upwelling, the upward nutrient supply, and the resulting biological productivity. Section 2 focusses on equatorial upwelling, section 3 on the Gulf of Guinea coastal upwelling and section 4 on the tropical Angolan upwelling. In section 5, we discuss longer-term changes and global warming and its relation to the seasonal cycle of upwelling, and, finally, in section 6, we provide a conclusion and outlook.

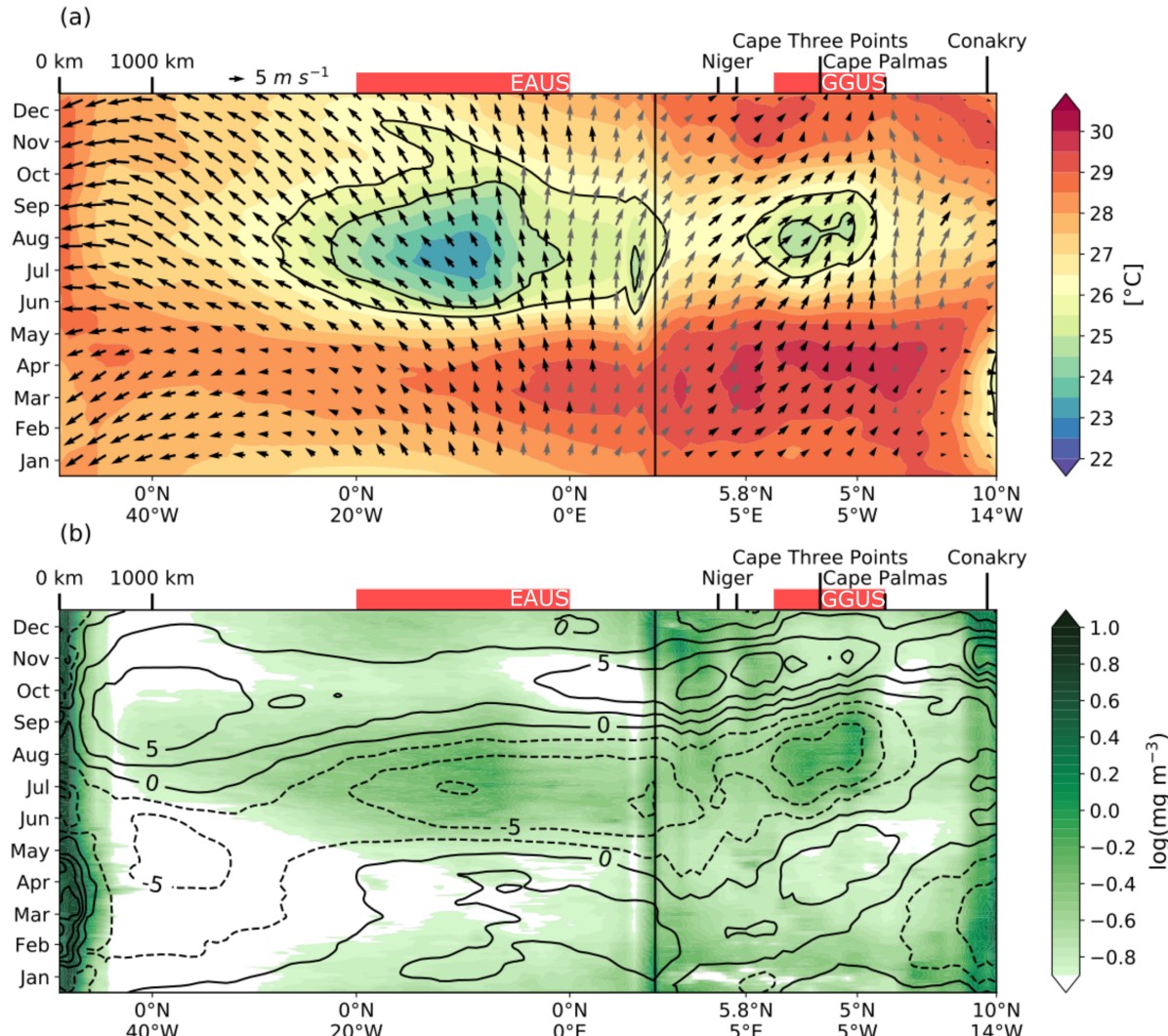

**Fig. 3** Seasonal cycle of (a) sea surface temperature (shading) and wind stress (arrows) and (b) chlorophyll concentration (shading) and sea level anomaly (contour lines, unit is cm) along the equatorial (left of the vertical black lines) and Gulf of Guinea coastal waveguides (right of the vertical black lines). Marks at the lower x-axis give geographic coordinates and marks at the upper x-axis give a scale for the distance and geographic locations along the waveguides. Also included in (a) are contours of the 25°C and 26°C isotherms to highlight equatorial and coastal upwelling. Upwelling- and downwelling-favourable winds in (a) are marked by black and grey arrows, respectively. Upwelling-favourable winds have an eastward component along the equatorial waveguide and an alongshore-equatorward component along the coastal waveguide. Positive and negative sea level anomaly in (b) are marked by solid and dashed contour lines, respectively; mean sea level anomaly is removed. SST data are from the Microwave OI SST product, wind data are from CCMP, chlorophyll data are from Copernicus-GlobColour, sea level anomaly data are from Copernicus DUCAS. All data is averaged for 1998-2020 within a 1° band along the equator (±0.5°) and within 1° distance along the coast.

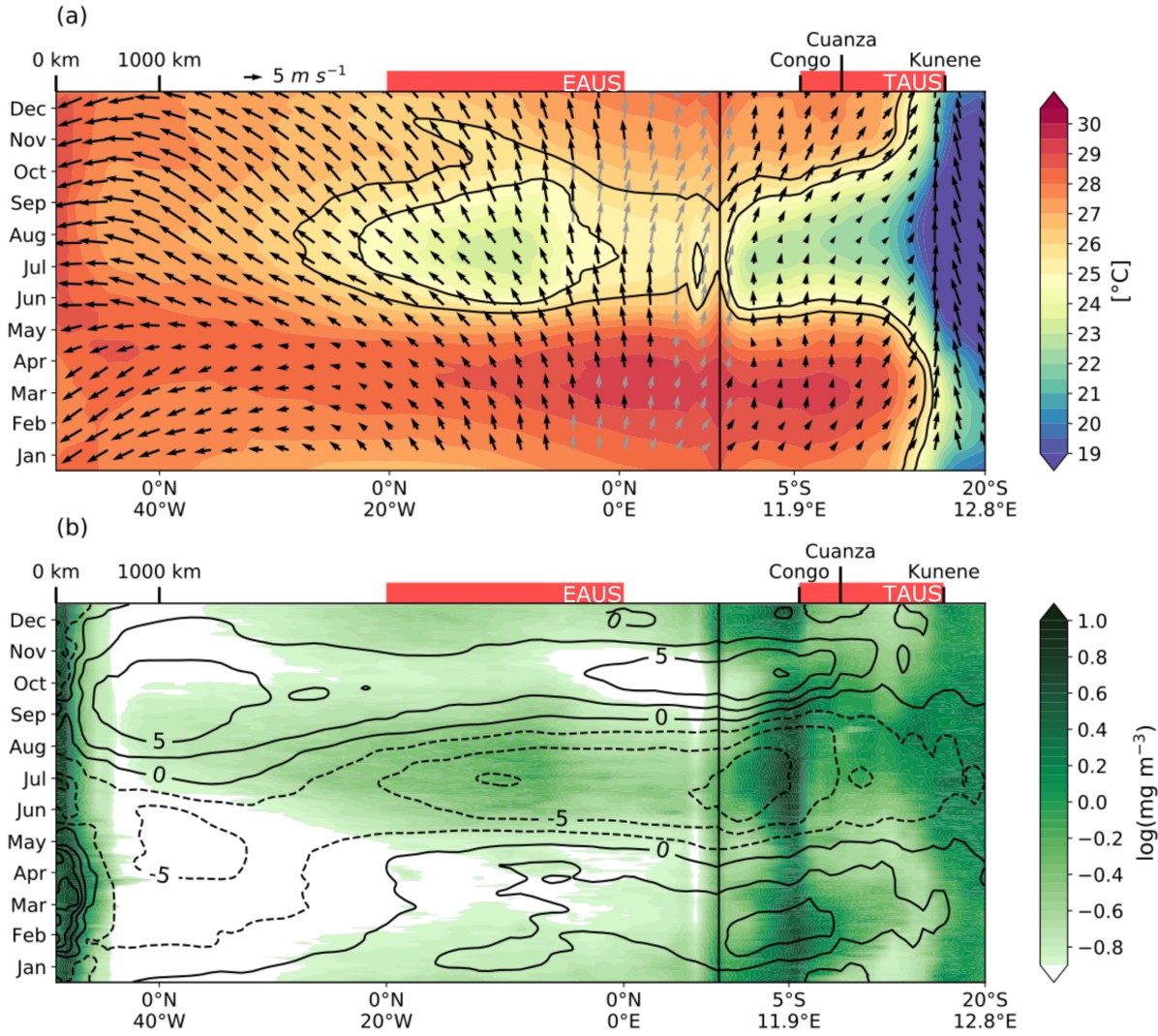

**Fig. 4** Same as Fig. 3, but along the southwest African coastal waveguide (right of the vertical black lines).

## 2 Equatorial upwelling

The equatorial upwelling transports cool, nutrient-rich waters toward the surface of the equatorial Atlantic. Its influence at the surface is more pronounced in the eastern part of the basin, where it gives rise to the development of the ACT. Its intensity is modulated by a seasonal cycle composed of an annual and a semiannual component, with a primary SST minimum in July-August and a secondary minimum in November-December (Fig. 3) (Okumura and Xie, 2006; Jouanno et al., 2011a). First insights into the seasonal evolution were obtained from observational studies in the 1980s, revealing a close link between seasonal surface cooling and vertical movements of the thermocline (Merle, 1980; Voituriez et al., 1982). Indeed, periods of surface cooling in the eastern equatorial Atlantic are in phase with the thermocline upwelling in May-June and November (Fig. 5). By using forced ocean models, Philander and Pacanowski (1981) have revealed variations of the equatorial thermocline as forced by the seasonal cycle of the wind stress: stronger easterlies result in a stronger uplift of the thermocline in the eastern equatorial Atlantic. These authors discussed the response of the Atlantic Ocean to the seasonal wind forcing as an equilibrium response that can be understood as a succession of steady states. However, such an equilibrium response requires the dominance of low baroclinic mode equatorial Kelvin and Rossby waves that propagate fast enough to adjust the thermocline to the wind forcing within the seasonal cycle (Ding et al., 2009; Hormann and Brandt, 2009; Philander and Pacanowski, 1986, 1981). Beside the eastern thermocline uplift, the equatorial easterlies

force a strong eastward thermocline flow, the EUC, that supplies the upwelling in the eastern
equatorial Atlantic (Johns et al., 2014; Schott et al., 1998).
The equatorial upwelling is an integral part of the STCs that are driven by the easterlies away
from the equator. The meridional divergence calculated from the Ekman transports at about
10°S and 10°N is about 20 Sv (Schott et al., 2004; Tuchen et al., 2019). Easterlies at the equator
additionally result in equatorial upwelling that is part of the tropical cells (Perez et al., 2014).
Tropical cells are similar overturning circulations as the STCs, but confined only to the upper
100 m with upwelling at the equator and downwelling at latitudes of about $\pm$3-5°. The annual
mean tropical cells in the central tropical Atlantic are found to be asymmetric with respect to
the equator; the northern cell extends into the southern hemisphere. This behaviour can be
explained by the presence of southerly winds peaking during boreal autumn that drive a cross-
equatorial northward surface flow at the equator (Heukamp et al., 2022). This circulation
feature, often referred to as the equatorial roll, has maximum southward return flow at about 50
m depth and upwelling and downwelling slightly south and north of the equator, respectively.
The upwelling velocity in the equatorial Atlantic is often estimated from local wind forcing as
the sum of the Ekman pumping due to the zonal wind stress, meridional wind stress, wind stress
divergence and wind stress curl (Caniaux et al., 2011). By using a realistic model of the
equatorial Atlantic, particularly including the full dynamic response to the wind forcing,
Giordani and Caniaux (2011) show that the dominant term driving the equatorial upwelling is
still the forcing by zonal wind stress. The importance of the forcing by the wind stress
divergence and the wind stress curl is, however, overestimated and underestimated,
respectively, in the Ekman theory compared to the applied dynamic model.
Over the past decade, several studies have revealed that turbulent mixing is the strongest
cooling term of the mixed layer heat budget during the onset of the ACT and sets the spatial
distribution and temporal variability of equatorial surface cooling (e.g., Jouanno et al., 2011a).
Turbulent mixing at the base of the mixed layer that drives heat flux out of the mixed layer into
the deeper ocean is dominantly induced by the vertical shear of the zonal equatorial currents,
that is the westward South Equatorial Current at the surface and the eastward EUC at the
thermocline level (Hummels et al., 2013). Different processes such as the seasonal variability
in strength and core depth of the zonal currents, vertical shear associated with intraseasonal
waves, the seasonally varying meridional circulation, and the deep-cycle turbulence contribute
to the spatial and temporal variability of equatorial mixing (Moum et al., 2022; Heukamp et al.,
2022). Using the diapycnal heat flux derived from observations, the seasonal mixed layer heat
budget at the equator could be closed to a large extent and the seasonal development of the
mixed layer temperatures reasonably well explained (Hummels et al., 2014).
An important consequence of upwelling is the increase in biological productivity that is
primarily dependent on nitrate supply (Herbland and Voituriez, 1979; Loukos and Memery,
1999; Radenac et al., 2020; Moore et al., 2004). There is a strong similarity between the
seasonal cycles of phytoplankton concentration and SST in the cold tongue area (Fig. 3)
(Jouanno et al., 2011b), suggesting that the same physical processes control the downward heat
flux out of the mixed layer and the upward supply of nitrate to the euphotic layer. This was
confirmed by the analysis of repeated sections of PIRATA service cruises and outputs from a
coupled physical-biogeochemical model (Radenac et al., 2020). Surface chlorophyll
concentrations in the ACT peak in July-August and exhibit a secondary maximum in
December-January (Figs. 3b and 4b, Fig. 5c). Radenac et al. (2020) showed that the primary
phytoplankton bloom in July-August is due to a strong vertical nitrate input to the equatorial
euphotic layer in May-July, and the secondary bloom in December is due to a shorter, moderate
input in November-December (Fig. 5d). Their analysis of the nitrate balance in the upper ocean
suggests that vertical advection controls the vertical movement of the nitracline and that vertical
diffusion allows nitrate to reach the mixed layer (Figs. 5e, f). However, already noted by
Monger et al. (1997), the phytoplankton concentration levels remain high beyond the primary
bloom period in July-August, despite the fact that the thermocline/nitracline has dropped back
to pre-uplift depth in September. Radenac et al. (2020) pointed towards a different behaviour
of the EUC during boreal spring and autumn, where a shallow EUC during boreal spring might
prevent upward mixing of nitrate compared to the deep phase of the EUC during boreal autumn,
when nitrate more easily reaches into the shear zone above the EUC core (Fig. 5). However,
also the equatorial role being at maximum strength during boreal autumn (Heukamp et al.,
2022) might contribute to the nitrate supply into the mixed layer by upwelling slightly south of
the equator.

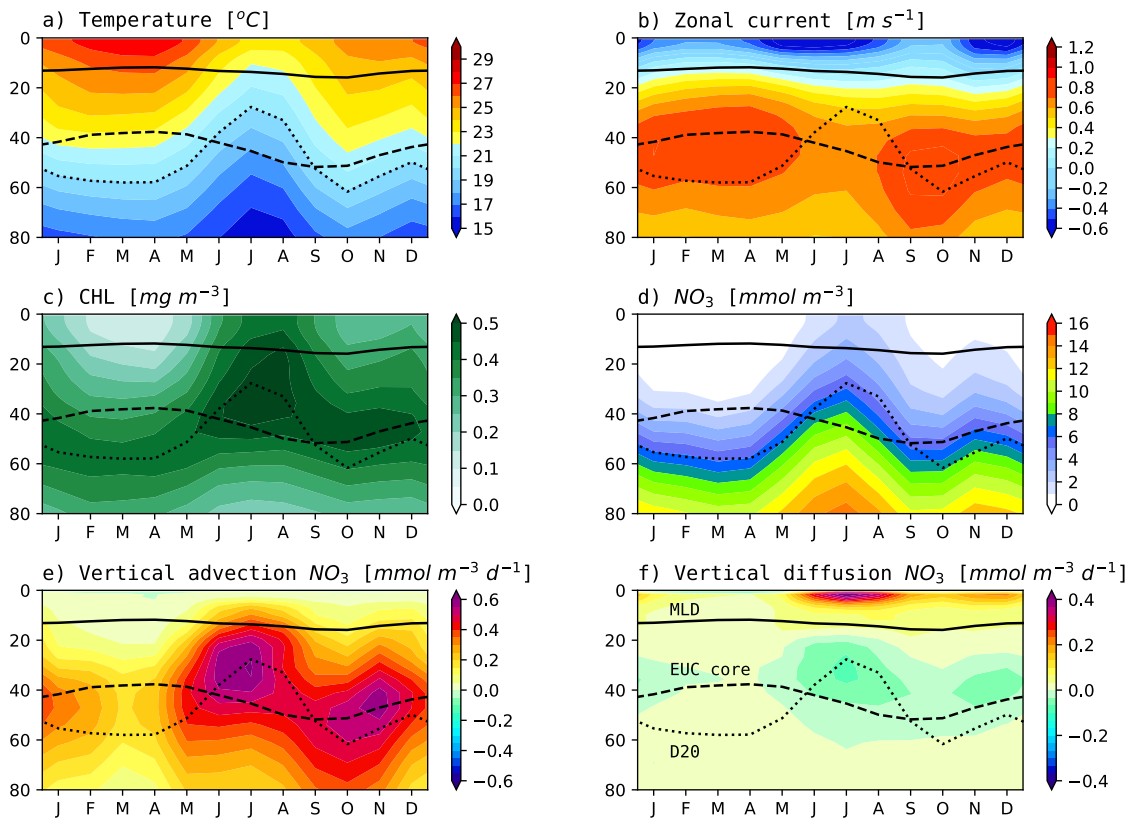

**Fig. 5** Seasonal cycle of vertical profiles of (a) temperature, (b) zonal velocity, (c) chlorophyll, (d) nitrate, (e) vertical advection, and (f) vertical diffusion horizontally averaged in 1.5°S–0.5°N, 20–5°W. The depths of the mixed layer (upper solid line), of the EUC core (dashed line), and of the 20°C isotherm (dotted line) are indicated. Model output is taken from Radenac et al. (2020).

Analysis of the PIRATA shipboard sections and model outputs in Radenac et al. (2020) showed
that waters transported eastward by the EUC have in general relatively low nitrate
concentrations compared to nearby water bodies to the north and south. This is most likely due
to the source waters of the EUC that arrive from the oligotrophic layers of the subtropical South
Atlantic (Schott et al., 1998; Johns et al., 2014; Tuchen et al., 2022a). The model simulations
by Radenac et al. (2020) also revealed that the EUC core does not follow the thermocline depth
(as defined as the 20°C isotherm, Fig. 5). While in the eastern equatorial Atlantic, the vertical
migration of the thermocline undergoes a semiannual cycle in accordance with the local wind
forcing, the EUC core depth has a dominant annual cycle (Brandt et al., 2014). This non-
equilibrium response of the equatorial Atlantic to the seasonal wind forcing could be explained
by resonant equatorial basin modes composed of eastward and westward propagating equatorial
Kelvin and Rossby waves, respectively (Brandt et al., 2016). As the thermocline depth is a good
proxy of the nitracline (Fig. 5d), the EUC transports elevated nitrate and phytoplankton
concentrations when the EUC core is close to or deeper than the thermocline or nitracline,
which is the case during July-August and to a lesser extent in December (Fig. 5).

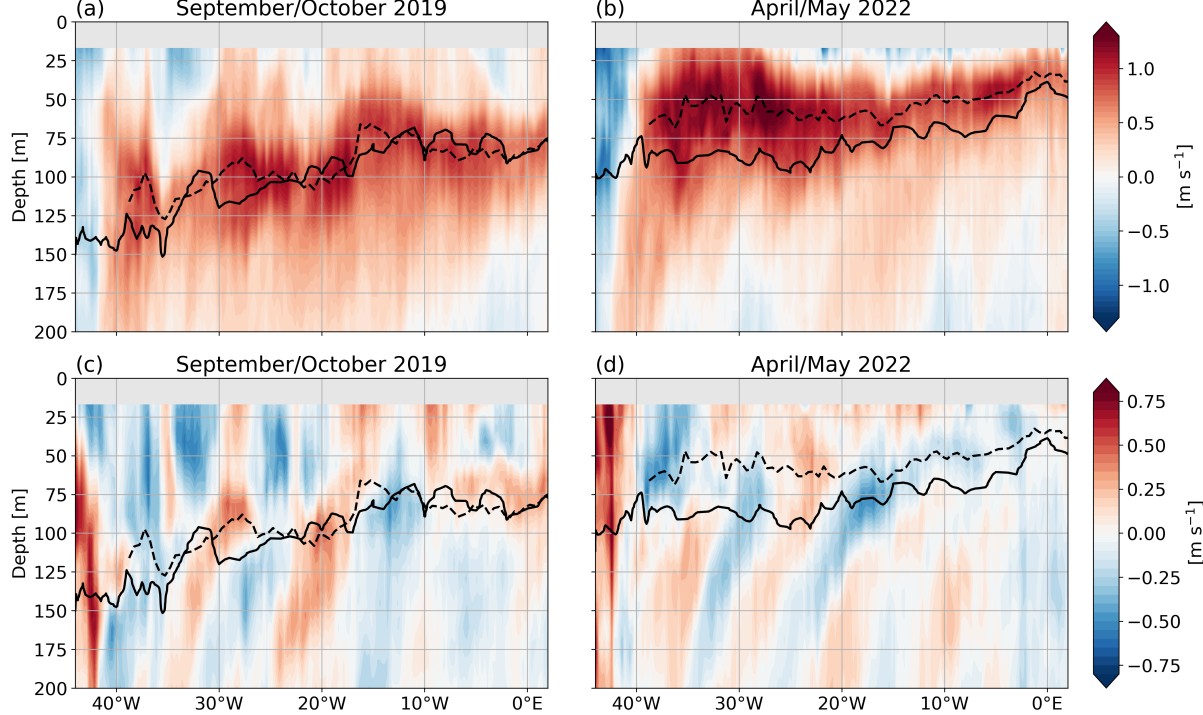

**Fig. 6** Zonal (a, b) and meridional (c, d) velocity measured along the equator in September-October 2019 (a, c)
and in April-May 2022 (b, d). Note, the different colour scales for the zonal and meridional velocities. EUC core
depth is marked by dashed lines and the 20°C isotherm (as a proxy of the thermocline and the nitracline) by the
solid lines.

Measurements along the equator during two cruises in boreal autumn (Fig. 6a) and boreal spring
(Fig. 6b) reveal the basin-wide character of the up- and downward movement of the EUC core
relative to the thermocline depths. During boreal autumn, the EUC core closely follows the
thermocline, while during boreal spring, it is located clearly above the thermocline. This
behaviour can be associated with the resonance of the equatorial basin at the annual period. The
period of a resonant equatorial basin mode is given by the total travel time of an equatorial
Kelvin wave and its reflected equatorial Rossby wave. For the width of the equatorial Atlantic
basin, the resonance period of the 4[th] baroclinic mode is close to the annual cycle (Brandt et al.,
2016). This basin mode is associated with maximum eastward velocity in the near-surface layer
in boreal spring and maximum westward flow in boreal autumn. A specific consequence of the
relative movement of EUC core and thermocline depths is that the thermocline and thus the
nitracline during part of the year vertically migrates into the shear zone above the EUC core.
As the upper shear zone of the EUC supports strongly elevated turbulent mixing (Hummels et
al., 2014; Hummels et al., 2013; Jouanno et al., 2011b; Moum et al., 2022), enhanced upward
nutrient flux occurs during those periods. Such behaviour was identified in the model study of
Radenac et al. (2020) showing a maximum in the near-surface diffusive nitrate flux into the
mixed layer in July-August and a secondary maximum in November-December (Fig. 5f).
Beside a seasonal cycle, the productivity on the equator shows elevated intraseasonal and
shorter-term variability. In particular, tropical instability waves (TIWs) and wind-forced
intraseasonal waves play an important role in stimulating locally productivity due to both
meridional advection of nitrate and chlorophyll as well as due to events of enhanced vertical
advection and mixing (Athie and Marin, 2008; Menkes et al., 2002; Jouanno et al., 2013). TIWs
are found to be associated with strong mixing events (Moum et al., 2009) or the generation of
fronts (Warner et al., 2018) that both can drive upward nutrient supply. Resulting high
productivity events could be observed during the boreal autumn cruise (Figs. 6a, c) that took
place shortly after the seasonal maximum of the TIW activity (Sherman et al., 2022).

## 3 Gulf of Guinea upwelling


In the Gulf of Guinea, coastal upwelling occurs seasonally along the northern coast, between
10°W and 5°E, from Côte d'Ivoire to Nigeria (Hardman-Mountford and McGlade, 2003). It
plays a key role in primary production and local fisheries and is therefore of large socio-
economic importance for the bordering countries (Koné et al., 2017; Amemou et al., 2020).
SST variability in the GGUS is suggested to modulate the amplitude of the African monsoon
and thus has influence on regional climate (Caniaux et al., 2011; Djakouré et al., 2017). The
GGUS is composed of two main upwelling cells, an eastern cell east of Cape Three Points
(4°44'N, 2°05'W) and a western cell east of Cape Palmas (4°22'N, 7°43'W), that are marked
by regions of reduced SST near the coast in satellite data (Fig. 3a) (Wiafe and Nyadjro, 2015).
Different physical processes have been proposed to explain the presence of the coastal
upwelling in the GGUS. In early studies, the coastal upwelling has been related to the
strengthening of the geostrophic coastal current by local and remote wind forcing. Indeed, the
seasonal strengthening in the eastward-flowing Guinea Current contributes to enhance the
meridional tilt of the thermocline, thereby bringing cooler subsurface waters near the coast
closer to the surface (Colin et al., 1993; Ingham, 1970; Bakun, 1978; Philander, 1979). The link
between SST and wind stress curl in the GGUS was first suggested by Katz and Garzoli (1982)
and Garzoli and Katz (1983). By using a model of the tropical Atlantic, Philander and
Pacanowski (1986) showed the influence of both wind components and the wind stress curl on
the upwelling in the GGUS. Additionally, Marchal and Picaut (1977) analysed isotherm
displacements between Ghana and Côte d'Ivoire and suggested that vertical pumping by
cyclonic eddies generated downstream of Cape Three Points and Cape Palmas could explain
upwelling of cool waters. However, modelling results by Djakouré et al. (2014) did not confirm
that the cyclonic eddies generated downstream of the capes contribute to the upwelling. Instead,
Djakouré et al. (2014) and Djakouré et al. (2017) suggested that the upwelling downstream of
Cape Palmas is associated with the nonlinear dynamics of the Guinea Current. The inclusion of
the nonlinearity in the momentum equations of their model results in an inertial detachment of
the Guinea Current from the coast after passing Cape Palmas. The geostrophic adjustment at
the coastward flank of the current then leads to thermocline upwelling downstream of Cape
Palmas. It is worth noting that the thermocline depth, the strength of the coastal current, and
thus the upwelling, are all under the seasonal remote influence of the equatorial ocean through
the propagation of equatorial Kelvin and Rossby waves as well as CTWs (Moore et al., 1978;
Clarke, 1979; Servain et al., 1982; Picaut, 1983; Adamec and Obrien, 1978). Such remote
influence is also indicated by the seasonal cycle of the sea level anomaly along the equatorial
and coastal waveguides (Fig. 3b) and was found for intraseasonal wave propagations as well
(Polo et al., 2008; Imbol Koungue and Brandt, 2021).
By using a model of the tropical Atlantic with an embedded high-resolution nest for the Gulf
of Guinea, Djakouré et al. (2014) and Djakouré et al. (2017) performed sensitivity experiments
to identify the dominant processes driving the seasonal upwelling in GGUS. The sensitivity
includes experiments with a changed coastline without the capes (Djakouré et al., 2014) and
with the nonlinear terms in the momentum equations responsible for the advection of
momentum removed (Djakouré et al., 2017). The spatial distribution of the mean SST for the
major upwelling season (July-September) is shown for their reference simulation in Fig. 7. A
comparison of the sensitivity experiments with the reference simulations (Fig. 8) shows that the
sea surface during boreal summer is still colder than the 25°C (chosen as a threshold for the
presence of coastal upwelling), when the capes are removed (Fig. 8b). The western upwelling

cell disappears only when the nonlinear terms in the momentum equations are removed and the Guinea Current is trapped at the coast (Fig. 8c).

The thermocline depth, superimposed on the SST in Fig. 8, is in each of these configurations closest to the surface during the upwelling season. During this period, in the simulation without capes, the thermocline has a structure almost identical to that of the realistic configuration. However, the thermocline depth is always larger than 20 m in the simulation without capes and thus deeper than in the reference simulation. In the simulation without nonlinear terms, the deepening of the thermocline relative to the reference simulation is stronger in the western upwelling cell than in the eastern upwelling cell (west and east of Cape Three Points, respectively) resulting in strongly reduced cooling in the western upwelling cell when advection of momentum is removed. The sensitive experiments demonstrated that advection of momentum is the main contributor to the vertical pumping of the western upwelling cell; the cooling of the eastern upwelling cell is mainly associated with the offshore Ekman transport (Djakouré et al., 2017).

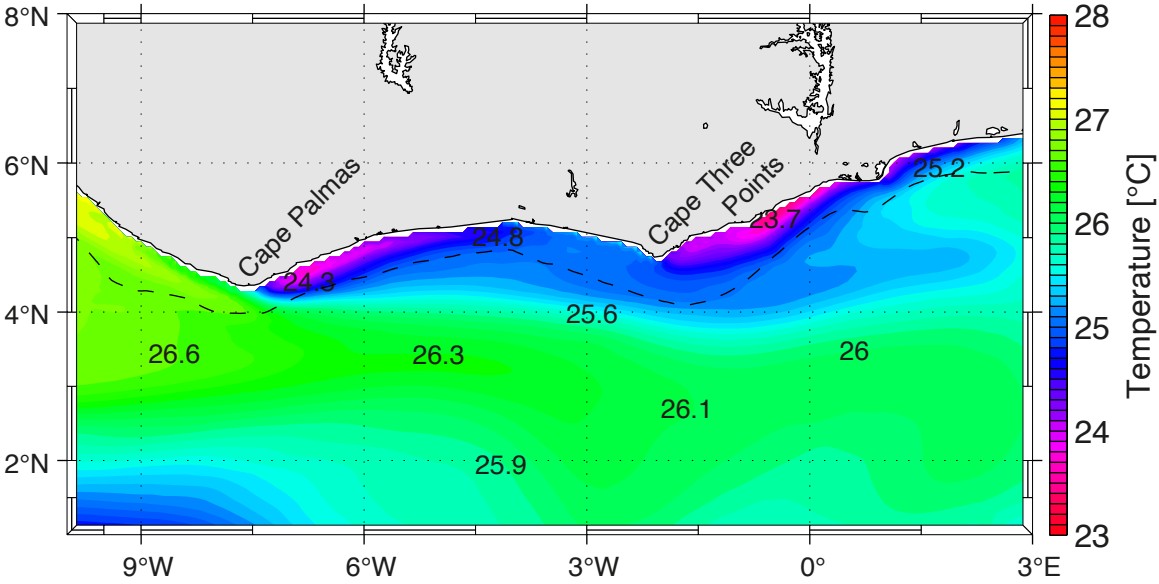

**Fig. 7** Mean SST (°C) for the major cold season (July - September) of the reference experiment by Djakouré et al. (2017). The dashed line represents the 1000 m isobath. Model output is taken from Djakouré et al. (2017).

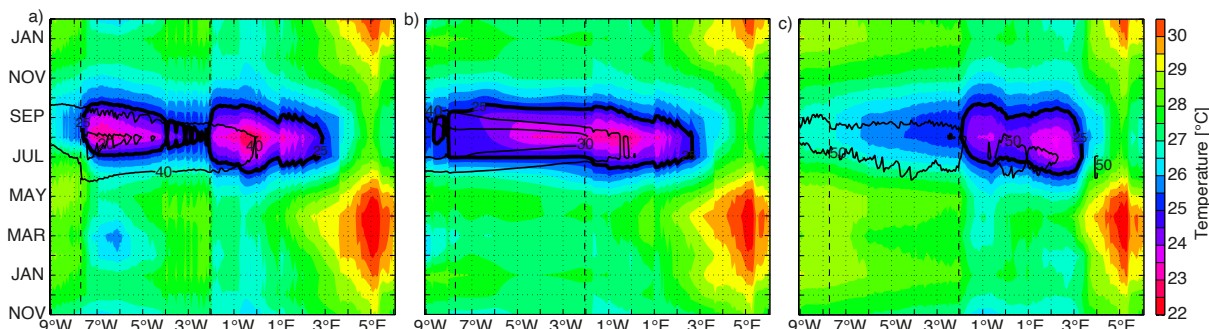

**Fig. 8** Hovmöller diagrams of SST (°C) along the coast of the GGUS from 9°W to 6°E of (a) the reference experiment, the idealized experiments (b) without capes and (c) without inertial terms. The thermocline depth (20°C isotherm) is superimposed (thin contour lines, unit is m). The vertical dashed black lines represent the longitude of Cape Palmas and Cape Three Points, see Fig. 7. The time axis in months extends from November to February of the following year. The 25°C isotherm is additionally marked to highlight the coastal upwelling (thick contour lines). Model output is taken from Djakouré et al. (2014) and Djakouré et al. (2017).

In the eastern part of the GGUS that is dominantly wind-driven (Fig. 3a), coastal cooling weakens toward the east while approaching the Niger River mouth (Fig. 9). Earlier studies have shown that onshore geostrophic flow can compensate wind-driven offshore transport, thus reducing upwelling in some regions (Marchesiello and Estrade, 2010; Rossi et al., 2013). Using a realistic regional model configuration, Ekman and geostrophic coastal upwelling indices were compared to coastal vertical velocities along the northern Gulf of Guinea coast, during the boreal summer season (Alory et al., 2021). Indeed, the upwelling indices were able to explain a large part of vertical velocity variations along the coast. They also showed that wind-forced coastal upwelling is reduced by about 50% due to onshore geostrophic flow east of 1°E (Fig. 9a). Note that the Ekman coastal upwelling index shown in Fig. 9a only takes into account the Ekman transport, as Ekman pumping has little influence in this region (Wiafe and Nyadjro, 2015). The onshore geostrophic flow is associated with a sea level slope increasing toward the east. It is driven by density differences along the coast, from relatively cool and salty waters in the upwelling core east of Cape Three Points (Fig. 7) to warm and fresh waters in the Niger River plume and largely compensates offshore Ekman transport and therefore reduces upwelling (Fig. 9a).

The comparison of a reference simulation with a simulation in which river run-off is removed, revealed that the Niger River discharge contributes to induce an onshore geostrophic surface flow, but additionally causes a thinning of the mixed layer. Overall, there is no net effect of the river discharge on the near-surface geostrophic transport from which the geostrophic upwelling index is derived. Nevertheless, the Niger River discharge induces a coastal warming reaching 1°C near 2°E (Fig. 9b), which likely is the result of reduced turbulent mixing by the enhanced salt stratification (Alory et al., 2021). The summer upwelling season corresponds both to a maximum thinning of the mixed layer and maximum surface chlorophyll concentration along the coast (Toualy et al., 2022). Riverine nutrient inputs may be more or less compensated by a reduced upward nutrient flux due to discharge-driven increased stratification as there is no strong chlorophyll signal in the plume region (Fig. 1).

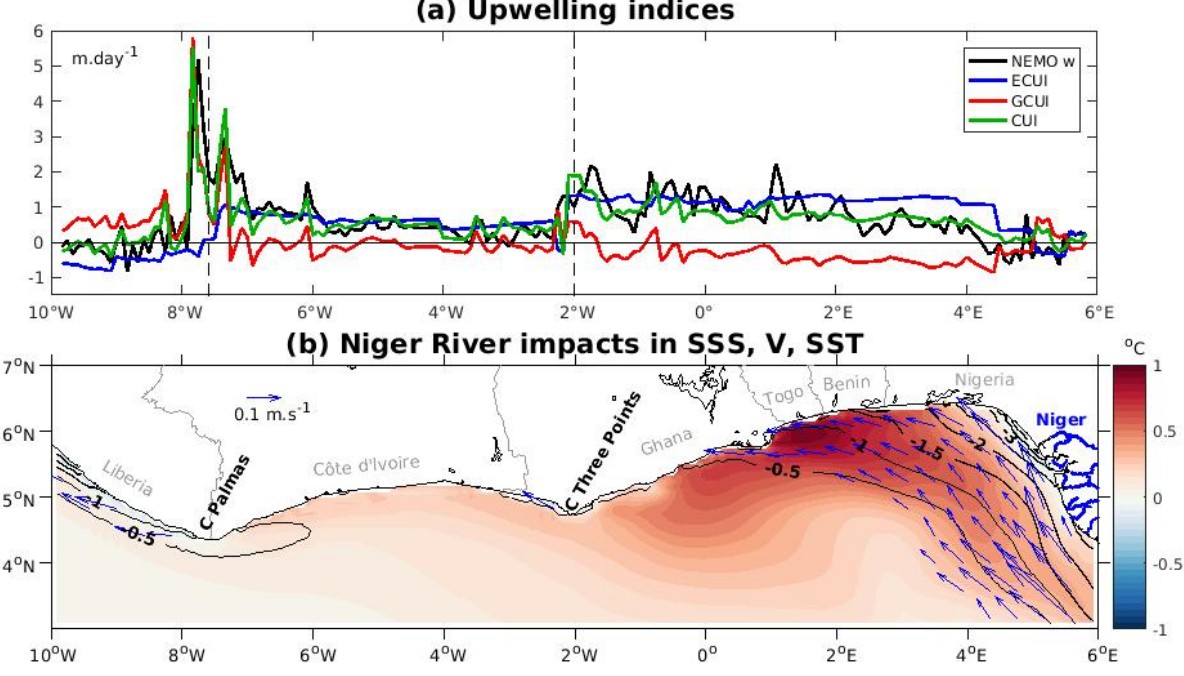

**Fig. 9** (a) Climatological mean (2010-2017) boreal summer (July - September) coastal upwelling index (CUI, green line), defined as the sum of Ekman (ECUI, blue line) and geostrophic (GCUI, red lines) coastal upwelling indices, compared with coastal vertical velocity at the base of the mixed layer in the reference NEMO simulation (black line). Correlation between the CUI and vertical velocities is 0.72. Vertical dashed lines indicate the location

of Cape Palmas and Cape Three Points. (b) River effects on boreal summer sea surface salinity (contour lines), surface geostrophic current (arrows) and sea surface temperature (colour shadings), from a difference between the NEMO reference and a runoff-free simulation. Model output is taken from Alory et al. (2021).

## 4 Tropical Angolan upwelling

The Angolan waters host a highly productive ecosystem: the TAUS (Fig. 1). Located in the southern hemisphere between the Congo River mouth at 6°S and the Angola-Benguela frontal zone at about 17°S, the TAUS is of great socio-economic importance for local communities. Fishing supplies about 25% of the total animal protein intake of the Angolan population and is critical for economic security (Hutchings et al., 2009; Sowman and Cardoso, 2010; FAO, 2022). The productivity in the TAUS undergoes a distinct seasonal cycle (Fig. 4). In the TAUS during austral winter, maximum productivity is observed at the same time as the lowest SST and the strongest cross-shore temperature gradient are present (Tchipalanga et al., 2018; Awo et al., 2022; Körner et al., 2022). In contrast to other eastern boundary upwelling systems, the seasonality of the productivity in the TAUS cannot be explained by local wind forcing (Ostrowski et al., 2009). Prevailing southerly winds in the TAUS are generally weak throughout the year (Fig. 4a). Neither the seasonal cycle of alongshore wind stress nor of the wind stress curl are in phase with the seasonal cycle in productivity suggesting that other mechanisms drive the productivity seasonality in the TAUS (Körner et al., 2022).

One of the key dynamics modulating the TAUS on different time scales is the passage of CTWs (Bachèlery et al., 2016a; Kopte et al., 2018; Kopte et al., 2017; Illig et al., 2018b; Tchipalanga et al., 2018; Awo et al., 2022; Körner et al., 2022). CTWs that propagate poleward along the eastern boundary are forced remotely by wind fluctuations along the equator or locally by winds at the eastern boundary. Sea level satellite observations reveal the seasonal passage of four remotely forced CTWs throughout the year (Fig. 4b) (Rouault, 2012; Tchipalanga et al., 2018). A downwelling CTW marked by anomalously high sea level arrives at the Angolan coast in March followed by an upwelling CTW marked by anomalously low sea level in June/July. A secondary downwelling CTW propagates along the Angolan coast in October followed by a secondary upwelling CTW in December/January. The main component of the eastern boundary circulation in the TAUS is the poleward Angola Current (Kopte et al., 2017; Siegfried et al., 2019). Its variability is linked to equatorial ocean dynamics via the propagations of CTWs at different time scales (Kopte et al., 2018; Kopte et al., 2017; Imbol Koungue and Brandt, 2021). On seasonal time scales the poleward velocities of the Angola Current peak in October with a secondary maximum in February (Kopte et al., 2017).

The hydrographic conditions in the TAUS undergo distinct seasonal changes (Fig. 10). Conductivity, temperature, depth (CTD) data from fifteen years of biannual research cruises of the Nansen program (Tchipalanga et al., 2018) illustrate the seasonal differences between the primary downwelling phase in late austral summer and the primary upwelling phase in austral winter (Fig. 10). In late austral summer (February-April) the cross-shelf section derived from data averaged between 10°S and 12°S shows warm surface waters and a subsurface salinity maximum below low-salinity surface water. The subsurface salinity maximum is absent in austral winter (June-August). The isopycnals show evidence of down- and upwelling as they bend downward towards the shore in late austral summer and upward in austral winter. Furthermore, the isopycnals undergo a vertical displacement between the seasons (the 26.2 kg m$^{-3}$ isopycnal moves vertically by about 50 m). The vertical displacement of the permanent thermocline can be attributed to the passage of CTWs. The seasonal passage of four CTWs induces a semiannual cycle in the vertical isopycnal movements (Kopte et al., 2017; Rouault, 2012).

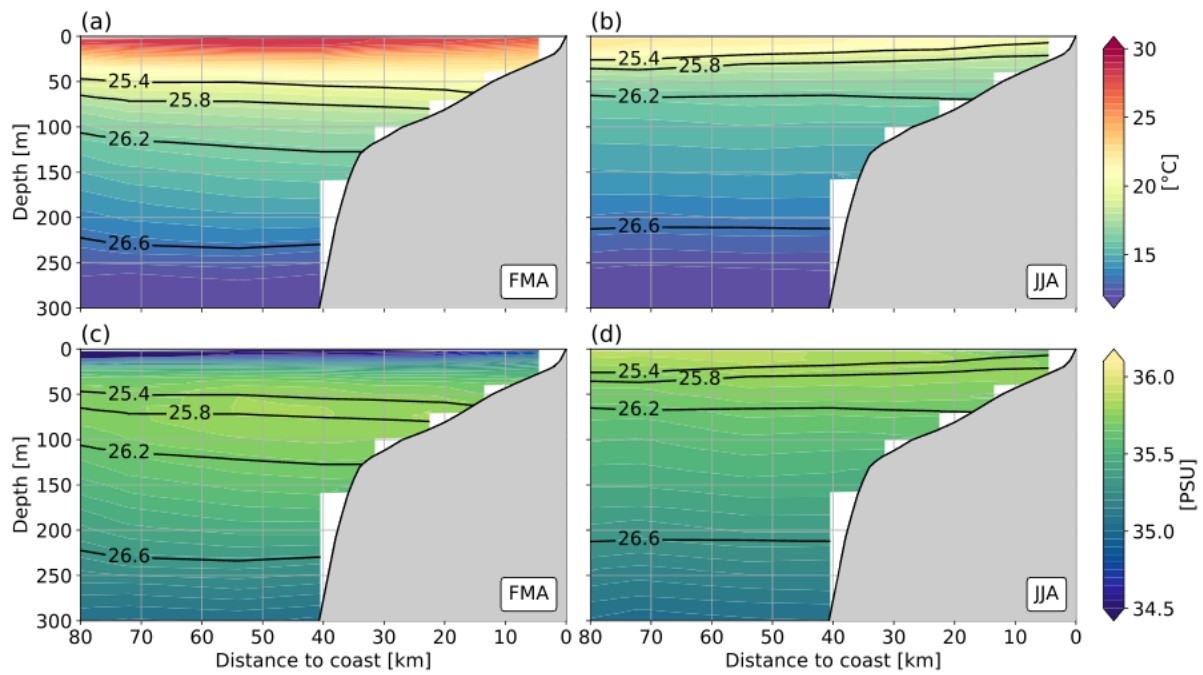

**Fig. 10** Hydrographic conditions between 10°S and 12°S during main downwelling phase, February-April (a,c) and main upwelling phase, June-August (b,d) inferred from the Nansen CTD dataset (Tchipalanga et al., 2018). CTD data is projected on mean topography (GEBCO) between 10°S and 12°S. Panels a and b show the temperature field, panels c and d the salinity field. Black contour lines mark potential density.

To understand the changes in SST in the TAUS, one has to account for other processes than the passage of CTWs. The SST which shows an annual cycle is dominantly driven by the surface heat fluxes. The advection of warm water by the Angola Current plays only a minor role (Körner et al., 2022). In the TAUS, SST is reduced in a narrow strip along the coast compared to further offshore (Fig. 2). The resulting negative cross-shore SST gradient has a semiannual cycle and is strongest between April and September, with a secondary maximum in December/January. The cross-shore SST gradient can neither be explained by surface heat fluxes which act to dampen the spatial SST differences nor by the weak horizontal heat advection. Ocean turbulence data revealed that turbulent mixing across the base of the mixed layer is strongest in shallow waters (water depths smaller than 75 m) and capable of setting up the negative cross-shore SST gradient. The semiannual cycle of the gradient can be explained by turbulent mixing acting upon seasonally different stratifications (Körner et al., 2022) as discussed below.

In contrast to SST, sea surface salinity (SSS) in the TAUS undergoes a semiannual cycle. Fresher water is found in the northern part of the TAUS in October/November and in February/March (Fig. 10) (Kopte et al., 2017; Lübbecke et al., 2019; Awo et al., 2022). An important source of freshwater in the TAUS is the Congo River discharge at 6°S, with a maximum discharge into the ocean in early December (Martins and Stammer, 2022). The observed freshwater in the TAUS is controlled by meridional advection via the Angola Current and peaks in phase with the strengthening of the Angola Current (Awo et al., 2022). Indeed, the Angola Current displaces the freshwater from the Congo River plume toward the TAUS, leading to elevated stratification with low-salinity water at the surface above a subsurface salinity maximum. This strong stratification favours the subsurface advection of high salinity water counteracting surface freshening via vertical salt advection and mixing at the base of the mixed layer (Awo et al., 2022).

Turbulent mixing is an important mechanism in the TAUS for the near-coastal cooling, upward salt flux, and upward nutrient supply (Awo et al., 2022; Ostrowski et al., 2009; Körner et al., 2022). Ocean turbulence data from six research cruises is used to analyse the distribution of vertical eddy diffusivity at a cross-shelf section at 11°S (Fig. 11). The vertical eddy diffusivity

is elevated near the bottom at the continental slope and shelf. Additionally, waters shallower than 75 m show enhanced diffusivities over nearly the whole water column. This finding suggests a dependence of mixing on bathymetry in the TAUS with stronger mixing occurring in shallow waters, similar to other upwelling systems (Schafstall et al., 2010; Perlin et al., 2005).

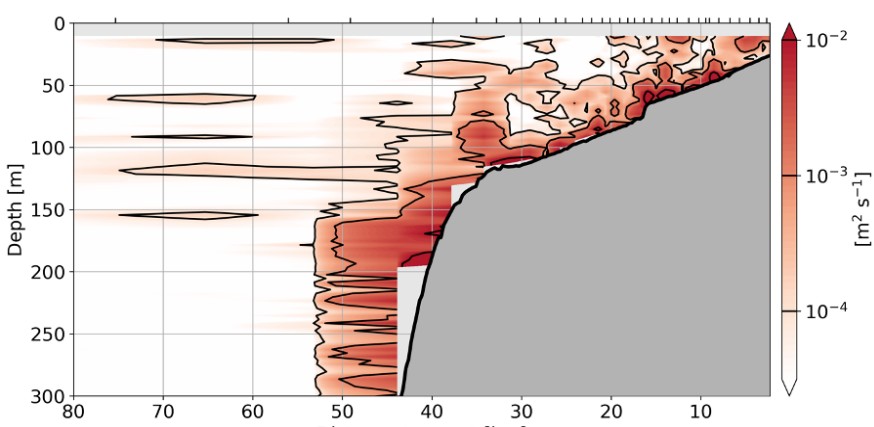

**Fig. 11** Vertical eddy diffusivity calculated from microstructure observation as a function of depth and distance to the coast. Measurements are taken at a section at about 11°S. Eddy diffusivity is calculated for each profile individually before profiles are binned together in groups of 20 profiles (black ticks on top mark the border of the 20-profile groups).

The elevated mixing rates in shallow waters of the TAUS can be explained by onshore propagating internal waves interacting with sloping topography. The main energy source of the internal wave field is assumed to be internal tides, which are generated by the interaction of the barotropic tide and the continental slope (Hall et al., 2013; Lamb, 2014). By applying a regional general circulation model forced solely by barotropic tides at the open boundaries, Zeng et al. (2021) found that in the TAUS a substantial part of the tidally generated internal wave energy propagates onshore and dissipates in shallow waters. Resulting enhanced near-shore mixing agrees well with observations. The seasonality of the spatially-averaged generation, onshore flux, and dissipation of internal tide energy is weak. This means that throughout the year, roughly the same amount of energy is available for mixing in shallow waters. However, the resulting mixing acts on seasonally different background stratifications that vary due to the passage of CTWs as well as due to surface heat and freshwater fluxes (Körner et al., 2022; Kopte et al., 2017). Zeng et al. (2021) showed that variations in the background stratification led to different effects of mixing on temperature: the sea surface in shallow waters near the coast is cooling stronger during phases of weak stratification than during phases of strong stratification.

The productivity season in the TAUS is in phase with the propagation of CTWs (Fig. 4). The chlorophyll concentration peaks around one month after the passage of the primary upwelling CTW in austral winter. Similarly, a secondary chlorophyll peak is visible after the passage of the secondary upwelling CTW in December/January (Figs. 1 and 4). However, the exact process of how the passage of the CTWs leads to an increase in primary production remains an open question. While the sea surface cooling depends on the background stratification (Zeng et al., 2021; Körner et al., 2022), the upward nutrient supply additionally depends on the background distribution of nutrients. An increased vertical nitrate gradient during phases of upwelling CTW in areas of high mixing would result in higher upward nitrate fluxes. Such changes in the background nitrate conditions associated with upward and onshore advection of nitrate during the passage of upwelling CTWs might be able to ultimately explain the seasonal productivity signals in the TAUS.

## 5 Relation between upwelling seasonality and longer-term variability

The seasonal upwelling in the three upwelling systems discussed here peaks approximately during the same period, i.e., in July-September (Fig. 1b-d). The area and season most impacted by the upwelling show marked interannual variability, whether in terms of SST (Keenlyside and Latif, 2007) or phytoplankton concentration (Chenillat et al., 2021). Fig. 12 shows the year-to-year variability of SST of the three tropical Atlantic upwelling systems averaged for the three months July-September. There are some similarities but also differences in the variability of the three upwelling systems. Outstanding is the most recent warm event in 2021 that peaks in all three upwelling systems.

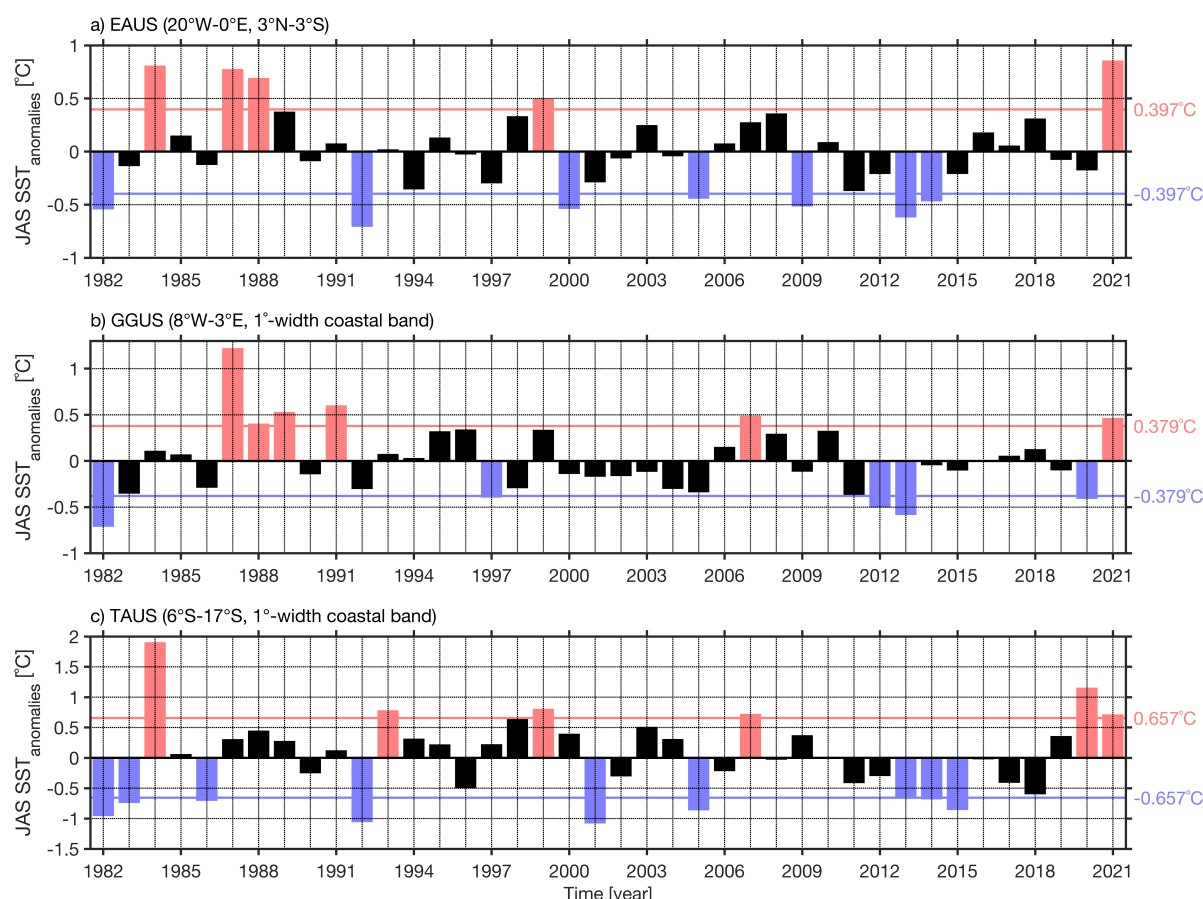

**Fig. 12** SST anomalies from 1982-2021 during the main upwelling season (July-September) averaged in the EAUS (a), GGUS (b), and TAUS (c). The red and blue rectangles highlight the extreme warm and cold events in the different regions, respectively. The horizontal red and blue lines show the standard deviation of the interannual SST anomalies during the main upwelling season (July-September). Anomalies are derived with respect to the seasonal cycle between 1982 and 2021 after subtracting the trend. SST data are from OI-SST (https://www.esrl.noaa.gov/psd/data/gridded/).

The dominant climate mode in the tropical Atlantic is the Atlantic Niño (Hisard, 1980; Ruiz-Barradas et al., 2000; Lübbecke et al., 2018). It is most pronounced in the equatorial cold tongue east of 23°W and peaks during June-July (Keenlyside and Latif, 2007). Anomalous warm or cold events are thus associated with anomalous deep or shallow thermocline and correspondingly with reduced or enhanced upwelling, respectively. Atlantic Niños and Niñas are associated with SSS variability as well (Awo et al., 2018) suggesting additional forcing of the equatorial and eastern boundary upwelling in the eastern tropical Atlantic as the coupling between subsurface and surface is reduced for enhanced near-surface stratification. During an Atlantic Niño, the southward shift of the ITCZ brings maximum rainfall in the eastern tropical Atlantic and potentially increases the flow of surrounding rivers, affecting near-surface stratification (Awo et al., 2018; Nyadjro et al., 2022). Besides the interannual variability,

decadal variability can impact the equatorial upwelling. Such variability might be associated with a changing strength of the STCs forced by off-equatorial easterlies (Rabe et al., 2008; Tuchen et al., 2020). Similarly, Brandt et al. (2021) found an intensification of the EUC for the period 2008-2018 that was linked to enhanced trade winds in the tropical North Atlantic likely associated with the Atlantic multidecadal variability (Knight et al., 2006). Other mechanisms that are suggested to impact the strength of equatorial cooling on decadal time scales include the decadal variability of TIWs. A decadal strengthening of TIWs was found to be associated with enhanced warming of the equatorial cold tongue by lateral eddy fluxes (Tuchen et al., 2022b). Coupled climate simulations also suggest the importance of surface heat fluxes in driving interannual to decadal cold tongue SST variability (Nnamchi et al., 2015). As discussed by Jouanno et al. (2017), such heat flux forcing is likely overemphasized due to large upper ocean temperature biases commonly found in climate models. Chang et al. (2008) analysed the impact of a changing Atlantic meridional overturning circulation (AMOC) on the tropical Atlantic on decadal to multidecadal timescales using simulations with a climate model. These simulations showed that a weakened AMOC results in a warmer equatorial Atlantic with reduced seasonal cycle and interannual variability. Similarly, a weakening of interannual variability is projected under a global warming scenario (Crespo et al., 2022; Yang et al., 2022). However, the impact of global warming or reduced AMOC on seasonal or interannual variability of productivity is highly uncertain as even the impact on upper ocean stratification is not coherent between different models and datasets, which is partly due to the fact that decadal trends in stratification or productivity are just emerging from the available observations (Roch et al., 2021; Sallee et al., 2021; Hammond et al., 2020).

In the GGUS, interannual variability is generally stronger in regions of strong seasonal variability as has been documented from satellite and in situ data (Wiafe and Nyadjro, 2015; Sohou et al., 2020). Potential drivers of interannual variability are similar to drivers of seasonal changes. They include changes in the wind forcing and turbulent mixing, and remote forcing associated with the Atlantic Niño (Jouanno et al., 2017; Wade et al., 2011). Processes involved in the 2012 cold anomalies in the GGUS, one of the coldest events observed over the last 30 years (Fig. 12b), have been investigated through a model heat budget (Da-Allada et al., 2021). Results revealed that the surface cooling at Cape Palmas was driven by changes in zonal advection and increased turbulent mixing due to a strengthening of the Guinea Current and associated vertical shear, while east at Cape Three Point, where seasonal upwelling is dominated by the wind forcing, it was driven by a strengthening of the zonal wind stress that increased the offshore Ekman transport.

In the TAUS, Benguela Niños and Niñas are the dominant modes of interannual climate variability (Shannon et al., 1986). Contrary to the variability in the EAUS and GGUS, interannual variability does not reach its seasonal maximum during the main upwelling season, but during the main downwelling season from March-May (Lübbecke et al., 2019). Still, some events are observed during austral winter such as the 1984 Benguela Niño (Imbol Koungue et al., 2019; Shannon et al., 1986). During a Benguela Niño or Niña, SST in the TAUS can be up to 2°C higher or lower than the climatology, respectively (Rouault et al., 2007; Rouault et al., 2018; Imbol Koungue et al., 2019; Imbol Koungue et al., 2021). These extreme events can have drastic consequences for the marine ecosystem (Gammelsrød et al., 1998) through modulations in coastal upwelling intensity, nutrients and oxygen content along the continental shelf (Bachèlery et al., 2016b). It can be assumed that forcing mechanisms of Benguela Niños and Niñas are similar to those of the seasonal upwelling variability. On the one hand, winds at the equator can generate equatorial Kelvin waves (Illig et al., 2004) that continue southward along the southwest African coast as CTWs and produce thermocline displacements along the eastern boundary (Polo et al., 2008; Imbol Koungue et al., 2017; Bachèlery et al., 2016a; Bachèlery et al., 2020). On the other hand, fluctuations of local alongshore winds (Richter et al., 2010) and other local processes such as freshwater inputs (Lübbecke et al., 2019) further generate SST

anomalies in the TAUS. For the satellite era, Prigent et al. (2020a) showed a weakening of interannual SST variability in the southeastern tropical Atlantic between 2000-2017 relative to 1982-1999. However, since 2018, two consecutive extreme coastal warm events have been recorded in the TAUS in 2019/2020 (Imbol Koungue et al., 2021) and in 2021 (Fig. 12c). The recent decades demonstrate a strong warming trend in the tropical Atlantic SST with the largest warming observed in the coastal upwelling regions off southwestern Africa including the TAUS (Tokinaga and Xie, 2011). Moreover, using observational data, Roch et al. (2021) discovered a change in upper-ocean stratification from subtropical to tropical conditions associated with a warming and freshening of the mixed layer between 2006 and 2020 in the southeastern tropical Atlantic (10°S-20°S; 5°W-15°E). Such changes in stratification are assumed to particularly impact the mixing-driving nitrate supply in the TAUS.

## 6 Conclusion and outlook

Here we have reviewed the physical processes in three major upwelling systems of the tropical Atlantic (10°N-20°S), the EAUS, the GGUS, and the TAUS, that drive the upwelling seasonality. Among them are the processes that locally impact the thermocline depth - often used as a proxy of the nitracline - such as zonal wind along the equator, alongshore wind in coastal upwelling regions, wind stress curl or the detachment of the boundary current. Remote processes associated with the propagation of equatorial Kelvin and CTWs affect the thermocline depth in the different upwelling regions on intraseasonal, seasonal and interannual timescales as well. The processes affecting the thermocline depth can be summarized under locally and remotely driven vertical advection which is able to transfer colder and nutrient-rich waters upward to the surface during active upwelling. Additionally, diffusive fluxes associated with turbulent mixing at the base of the mixed layer and within the thermocline transport heat downward and nutrients upward. While the dominant processes driving equatorial and coastal upwelling might be identified, we are only beginning to quantify their relative importance or to understand their interactions. Examples are the nonlinear interaction of locally and remotely forced boundary current variability and horizontal density anomalies or topography (Mosquera-Vasquez et al., 2014; Kämpf, 2007), the importance and characteristics of different CTW modes and their specific role in the vertical advection of nutrients (Bachèlery et al., 2020; Illig et al., 2018a; Illig et al., 2018b), or the role of intraseasonal variability and the eddy field in shaping the upwelling (Tuchen et al., 2022b; Thomsen et al., 2016). With the development of extremely high-resolution ocean models the importance of the mesoscale, submesoscale and their role in mixed layer dynamics and thermocline mixing emerged. Dedicated observational studies particularly in eastern boundary upwelling systems are required, focussing on these smaller-scale dynamics and aiming at understanding their impact on seasonal and longer-term changes of upwelling.

The EAUS can be characterized as a wind-driven upwelling system forced by different wind components at the equator and off the equator. Off-equatorial winds drive the STCs, which act on longer time-scales, mostly larger than 5 years (Schott et al., 2004; Tuchen et al., 2020). How their changes affect stratification and nutrient distribution is still an open question (Duteil et al., 2014). Wind changes along the equator generate upwelling and downwelling equatorial waves propagating along the equator and adjust the equatorial thermocline to reach an equilibrium with the wind forcing. Additional wave forcing originates from westward propagating Rossby waves and their reflection at the western boundary (Foltz and McPhaden, 2010a, b) or by CTWs generated at the western boundary (Hughes et al., 2019). A very specific response of an equatorial basin is the development of a basin resonance. Due to its width and the travel time of equatorial waves, the Atlantic basin is resonant at the $2^{nd}$ and $4^{th}$ baroclinic modes for the semi-annual and annual cycles, respectively. The resonance results in an EUC that vertically migrates largely independently of the thermocline (Brandt et al., 2016). During periods, when

the thermocline depth is shallower than the EUC core, turbulent mixing in the shear zone above the core of the EUC is essential for the downward heat flux and upward nitrate flux out and into the mixed layer, respectively (Jouanno et al., 2011b; Hummels et al., 2014). Similar resonances are found for the Indian Ocean (Han et al., 2011), while the Pacific Ocean due to its larger width develops resonances at lower baroclinic modes and/or larger periods.

In the GGUS different processes define the two upwelling centres, east of Cape Palmas and east of Cape Three Points. East of Cape Palmas, the inertial detachment of Guinea Current from the coast plays the most important role, while east of Cape Three Points, upwelling is mainly associated with the wind-forced coastal divergence (Djakouré et al., 2017). The upwelling in the TAUS, which is characterized by weak winds, is dominantly driven by a combination of remotely forced CTWs and turbulent mixing locally enhanced in shallow waters near the coast (Körner et al., 2022; Tchipalanga et al., 2018; Rouault, 2012).

Climate warming and change might impact the upwelling in the different regions and their seasonality (amplitude and phase) differently. Most obvious are probably future changes in the wind field, e.g., a strengthening of the winds in a warming world or poleward shifts of the main wind systems (Yang et al., 2020). Changes in the stratification and mixed layer depths are highly uncertain with recent studies suggesting an increase of the stratification at the base of the mixed layer together with a mixed layer deepening likely due to enhanced wind-driven upper-ocean turbulent mixing (Sallee et al., 2021; Roch et al., 2021). However, other processes such as lateral mixing, responsible for reducing nitrate concentrations in upwelling regions, or surface heat fluxes, might contribute as well. Recently, the multidecadal increase in the strength of TIWs and associated equatorward eddy heat flux was suggested to warm the EAUS (Tuchen et al., 2022b). Such eddy fluxes generally oppose the Ekman transport in upwelling systems. Air-sea heat and buoyancy fluxes were identified to modulate such compensation in idealized model simulations (Thomsen et al., 2021), suggesting that in a warming climate also changes in heat and freshwater fluxes have the potential to impact the upwelling strength via its impact on lateral eddy fluxes.

The identification of climate changes in upwelling systems is a major goal that requires the maintenance and further development of the tropical Atlantic observing system (Foltz et al., 2019). In particular, coastal upwelling regions show a sparse data coverage and the strengthening of the near-coastal observing system has a high priority. This requires a close cooperation with the coastal communities to jointly develop the research agenda according to collective interests and needs. Main questions regard the often-competing role of changes in wind forcing and stratification and the role of changing eddy fluxes and buoyancy forcing. What will be the consequences of changing upwelling amplitude and/or timing for biogeochemistry and biology? Overall, future upwelling studies require a close cooperation between different research disciplines focussing on the interaction between the physical, biogeochemical and biological systems and allowing an improved assessment of ecosystem management and fisheries.

**Data availability**

Publicly available datasets were used for this study. Chlorophyll data (1998-2020) are from the Copernicus-GlobColour dataset (https://doi.org/10.48670/moi-00281). The sea level anomaly data (1998-2020) were accessed via the Copernicus Server (https://doi.org/10.48670/moi-00148). Microwave OI SST and CCMP wind data (both 1998-2020) are available under https://www.remss.com. Also used are surface wind stress from ERA5 (https://cds.climate.copernicus.eu/). Hydrographic sections in Angolan waters have been produced using the Nansen CTD Dataset (https://doi.pangaea.de/10.1594/PANGAEA.887163).

**Author contribution**

PB outlined and wrote the manuscript. MK, RI, JJ, SD, GA produced the figures. All co-authors
contributed to and reviewed the manuscript.
**Appendix A**
List of abbreviations.
AMOC      Atlantic meridional overturning circulation
ACT      Atlantic cold tongue
CTD      conductivity, temperature, depth
CTW      coastally trapped wave
EAUS      equatorial Atlantic upwelling system
EUC      Equatorial Undercurrent
FAO      Food and Agriculture Organization
GGUS      Gulf of Guinea upwelling system
ITCZ      Intertropical Convergence Zone
TAUS      tropical Angolan upwelling system
TIWs      tropical instability waves
SSS      sea surface salinity
SST      sea surface temperature
STC      subtropical cells

**Acknowledgements**
The study was funded by EU H2020 under grant agreement 817578 TRIATLAS project. It was
further supported by the German Federal Ministry of Education and Research as part of the
BANINO (03F0795A) project, by the German Research Foundation through grant 511812462
(IM 218/1-1), and by the French Research Institute for Development (IRD) as part of the JEAI
IVOARE-UP project.

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
