# Peer review of "Physical processes and biological productivity in the upwelling regions of the tropical Atlantic"

_EGUsphere, 2022_

## Author Comment (AC1)

Authors response to comments by reviewer #1 to the manuscript "Physical processes in the upwelling regions of the tropical Atlantic" by Brandt et al. (pbrandt@geomar.de).
We would like to thank the reviewer for the detailed and helpful comments to improve the manuscript. Below, we use black text for the reviewer's comments and green text for our response.

RC1: 'Comment on egusphere-2022-1354', Anonymous Referee #1, 13 Jan 2023

General comments:

This is an interesting, extensive, and relevant compilation of knowledge about (upwelling in) the tropical Atlantic climate system. I find the introduction to be a bit disconnected which at times makes it hard to follow, the reader could be helped by working on the flow of the text and explaining why specific parts of the system are being introduced. Additionally, later on in the text statements are made that are not easily verifiable by the reader, e.g. wrt figure 3 and 4 and 5. More direction as to where the reader should focus and more explanation in the text would be helpful, especially to make this work accessible to a wider audience than the established tropical Atlantic community. Similarly the text at times mentions specific terms without either showing the equations or explaining what the terms represent. Either would be helpful here for completeness. Specific incidents are indicated below.

I recommend adding a connection to primary productivity / nutrient supply to the title, since it is discussed a lot in the text.

We added biological productivity to the title.

Specific comments

L50: add citation Yang, Yun, et al. "Suppressed Atlantic Niño/Niña variability under greenhouse warming." Nature Climate Change 12.9 (2022): 814-821.

Included the reference (Yang et al., 2022)

Ll 80-85 The areas GGUS tAUS and EAUS should be indicated in the figure, the focus here is said to be on inner upwelling and not coastal upwelling. Indication of the region would help the reader understand which areas are being discussed and which are not.

We now included the upwelling areas in Fig. 1, Fig. 3, and Fig.4. Areas are now defined in the caption and in the text.

Ll 88- 103 The discussion on where the water masses are coming from and going to : please add a sentence or two to the relevance of this discussion, potentially relating to primary productivity, OMZ, etc. Since this also is discussed in the individual sections (e.g. Ll243-245) this discussion here might be removed in favor of flow of the paper.

We reworked the section, removed the first sentence and added a sentence relating the supply of upwelling to oxygen.

L 104: connection between upwelling and ITCZ unclear, I recommend mentioning the relevance of this section to the upwelling in the tropical Atlantic in the beginning of this section

We now start that section as follows: The tropical Atlantic and its upwelling systems undergoes a strong seasonal cycle. Main driver are the seasonally changing winds associated with the meridional migration of the Intertropical Convergence Zone (Fig. 2).

Ll 133-135 please mark the upwelling favoring easterly winds in the figure, it is hard to see the variability in strength from Figure 1. When the winds are purely easterly they seem very weak (compared to September / October), should the reader focus solely on the region west of 40W? Please indicate in the figure and / or describe in the text.

We now mark upwelling-favourable winds in Figs. 3 and 4 by black arrows, downwelling-favourable winds by grey arrows. We also changed the text to clarify that for the EAUS easterly winds almost along the whole equator are discussed.

Ll 138-140 Is the region between Cuanza and Kunene meant here with particularly weak winds? The winds further south seem strong. Please indicate maybe with a different colour the arrows of the area under discussion, and / or add more description to the text.

It is the region between Congo and Kunene. The region is now marked in Fig. 4 to make this clearer. We now also mention the Kunene upwelling cell at the southern boundary of the TAUS that is part of the northern Benguela upwelling system when introducing the TAUS.

L 144 I am confused by the mention of inner tropical Atlantic upwelling, while the GGUS seems to follow the coast, similar to the tAUS. Again, indication of the areas in Fig 1 would help, and maybe a sentence on the differentiation between inner and coastal upwelling as it is used in this study.

With inner tropical Atlantic we referred to the region of the tropical Atlantic closer to the equator excluding part of the tropics close to the subtropics. This region includes the equatorial and coastal upwelling regions. However, we see that the term is not well defined and we removed "inner" completely, now always stating tropical Atlantic between 10°N and 20°S.

Ll 174-176 Which part of Figure 5 is indicated here? Maybe an extra figure with Thermocline movement in conjunction with temperature change should be shown here. Or alternatively Fig 3 / 4 are meant, using the SSH as proxy for thermocline? Related: the Figure caption of Figure 5 should mention the source of the data as do the other figures

We now included in Fig. 5 two new panels (temperature and zonal velocity). The seasonal temperature evolution (panel a) now shows the movement of the 20°C isotherm (as a proxy of the thermocline) as well as changes of the sea surface temperature. Furthermore, it is now mentioned that the model output is taken from Radenac et al. (2020).

Ll 188-201 Since the focus of this paper is seasonal variability, a note on what we (do not) know about the variability of the STCs and TCs as they relate to equatorial upwelling would be helpful.

We now include in the conclusion and outlook a short statement on STCs: Off-equatorial winds that drive the STCs acting on longer time-scales, mostly larger than 5 years (Schott et al., 2004; Tuchen et al., 2020). How their changes affect stratification and nutrient distribution is still an open question (Duteil et al., 2014).

L 200-202 "The different forcing terms.." it seems odd that in a review of the forcing terms of the tropical Atlantic upwelling the physical processes forcing the upwelling are only mentioned in a short sentence with a reference. I recommend expanding on this sentence and ideally drawing a connection to the next paragraph, turbulent mixing. Alternatively, a differentiation between the current work and Giordani and Caniaux 2011 would be helpful. Also since this review is on upwelling (and its impact on nutrient availability and primary production) the connection between upwelling and mixing could be explained.

We now expand on the wind forcing of the upwelling velocity as follows: The upwelling velocity in the equatorial Atlantic is often calculated from the wind forcing as the sum of the Ekman pumping due to the zonal wind stress, meridional wind stress, wind stress divergence and wind stress curl (Caniaux et al., 2011). By using a realistic model of the equatorial Atlantic particularly including the full dynamic response to the wind forcing, Giordani and Caniaux (2011) show that the dominant term driving the equatorial upwelling is still the forcing by zonal wind stress. The importance of the forcing by the wind stress divergence and the wind stress curl is, however, overestimated and underestimated, respectively, in the Ekman theory compared to the used model.

L 223 Fig 3 or 4 can be referenced in addition to figure 5 since they show the surface and 5 the column, might be more intuitive for the reader

Thank you is mentioned.

Ll223 Radenac reference, later on it is stated that the authors analysis PIRATA and models, please specify which dataset these results are based on as done later in e.g. L 241, 245

We now include in the caption to Fig. 5 a statement that the results are obtained from model output taken from Radenac et al. (2020).

L 255 Fig 5 ; EUC and 20C are shown in all panels

changed

L 277 December maximum is not clear in Fig 5d, looks similar throughout September - January. Fig 5c shows vertical advection maximum in November, how does this relate?

We mentioned in the text the near-surface diffusive nitrate flux that shows a maximum in July-August and a secondary maximum in November-December, which can be identified in Fig. 5f.

L 290 again confusion about inner vs coastal upwelling, explicit mention of coastal upwelling here (and throughout the text)

See above, "inner" is not used anymore.

Ll 296-297 I suggest indicating the cells in Figure 1

We now indicate the three upwelling systems in Fig. 1

Ll 313 "associated to the non-linear dynamics and its detachement.." Please add (half) a sentence on how this influences the upwelling

We added: The inclusion of the nonlinearity in the momentum equations of their model results in an inertial detachment of the Guinea Current from the coast after passing Cape Palmas. The geostrophic adjustment at the coastward flank of the current then leads to thermocline upwelling downstream of Cape Palmas.

Ll 325 what do these non-linear terms represent? In this overview being more specific about the physical process would be helpful

See point above regarding the nonlinearity in the momentum equations.

Ll 331-332 this is a bit more explicit "when the nonlinear terms are removed and the Guinea Current is trapped" but more explanation would again be helpful. Since this paper summarizes the physical processes behind upwelling it should be explicit about these processes.

We hope that the explanation regarding the role of the nonlinear terms for the inertial overshoot and the detachment of the Guinea Current clarified that point.

Ll 333-343 The discussion about the thermocline being closed to the surface in the simulation with least upwelling is difficult to follow. Earlier upwelling and upward movement of the thermocline have been positively correlated, how do they relate here? Seemingly the thermocline is shallower in the western upwelling cell while that cell has less upwelling (than the east), isn't this counterintuitive?

We clarify that statement as follows (it is the relative change of the thermocline depth in the sensitive experiment relative to the reference simulation that is important):
In the simulation without nonlinear terms, the deepening of the thermocline relative to the reference simulation is stronger in the western upwelling cell than in the eastern upwelling cell (west and east of Cape Three Points, respectively).

L 358 "that is mostly wind driven" can this be seen in Fig 3? It would be good to refer back to the (relevant section of that) figure

We now mark in Figs. 3 and 4 upwelling-favouring wind with black arrows and downwelling-favouring winds with grey arrows and reference Fig. 3a. However, the relatively stronger

wind forced upwelling east compared to west of Cape Three Points is shown best by the Ekman coastal upwelling index plotted in Fig. 9.

L 397 again please indicate the tAUS in Fig 1

See above, is included.

L 407 "are generally weak throughout the year" makes me think that it would also be good to indicate the tAUS region in Fig 4 or highlight the arrows in a different color (color coding arrows per upwelling zone might be a really good idea)

See above, is included.

Ll 415-416 "..four remotely forced CTWs throughout the year (Fig 4b)" can these be indicated in the figure, as arrows or similar

We think that it would overload the figure. However, we smoothed the somewhat noisy field in Fig. 4b to emphasize the phases of anomalously high and low sea level.

Ll 448-449 Indicate tAUS in figure 2? Is the very very narrow coastal strip e.g. in 4b meant here, or solely Fig 4c where the colder coastal SSTs seem more obvious? Again how do the authors distinguish between coastal and interior upwelling?

We now mention that SST is reduced in a narrow strip along the coast compared to further offshore (Fig. 2). We additional state that this region refers to water depths smaller than 75m.

Ll 451-455 description of coastal upwelling? It seems that the word inner in the beginning should be omitted or well defined.

It is omitted now.

Ll 489-490 "the spatially-averaged generation" of turbulence?

Should be clear (we don't see the misunderstanding): it is the spatially-averaged generation of internal tide energy

L 493 also evident in Figure 10?

In Fig. 10 mostly the isotherms upwelling (JJA) or downwelling (FMA) toward the coats are visible. The reduced SST near the coast is not the main point of that Figure.

L 496-497 related to increased mixing?

No, it is not related to mixing. As was written before: mixing acts on seasonally different background stratification. We try to make this clearer by adding:
However, the energy available for mixing acts on seasonally different background stratifications that varies due to the passage of CTWs as well as due to surface heat and

freshwater fluxes (Körner et al., 2022; Kopte et al., 2017). Zeng et al. (2021) showed that the variations in the background stratification lead to different effects of mixing on temperature: the sea surface in shallow waters near the coast is cooling stronger during phases of weak stratification than during phases of strong stratification.

Ll 502-504 what is the causal relationship here? More mixing = more cooling and therefore less stratification, but here the argument seems to be more mixing => less stratification => more cooling, can you be more explicit about the suggested series of events?

We hope that with the changes mentioned right above that this becomes clearer. The relationship is: less stratification => more effective mixing => more cooling. Here, we changed the text as follows:
While the sea surface cooling depends on the background stratification (Zeng et al., 2021; Körner et al., 2022), the upward nutrient supply additionally depends on the background distribution of nutrients.

L 514 suggest removing "it is"

Changed

L 522 additional forcing

Changed

Ll 522-524 this causality is not clear, please clarify

We changed the text as follows:
Atlantic Niños and Niñas are associated with SSS variability as well (Awo et al., 2018) suggesting additional forcing of the equatorial and eastern boundary upwelling in the eastern tropical Atlantic as the coupling between subsurface and surface is reduced for enhanced near-surface stratification. During an Atlantic Niño, the southward shift of Intertropical Convergence Zone (ITCZ) brings maximum rainfall in the eastern tropical Atlantic and potentially increases the flow of surrounding rivers, affecting near-surface stratification (Awo et al., 2018; Nyadjro et al., 2022).

L538 what is the timescale of the AMOC weakening? Decadal?

Included: on decadal to multidecadal timescales

L 540 add citation same as above Yun Yang

Done

Ll 544 "or productivity" maybe better to phrase "also indicated by trends in productivity"

That is not what we meant. We mean indeed stratification or productivity as two independent parameters for which decadal trends just emerging.

L 556 Please remind the reader how the influence of Ekman transport fits in with the seasonal modulation

We included the phrase: on decadal to multidecadal timescales

L 586 what does inner mean here

"inner" is removed. We now refer to "the tropical Atlantic (10°N-20°S)"

Minor:

Some inconsistencies with the plural and singular in the text, e.g. Ll 241-242 ..waters… has ..

Changed

**References**

Awo, F. M., Alory, G., Da-Allada, C. Y., Delcroix, T., Jouanno, J., Kestenare, E., and Baloitcha, E.: Sea Surface Salinity Signature of the Tropical Atlantic Interannual Climatic Modes, J Geophys Res-Oceans, 123, 7420-7437, https://doi.org/10.1029/2018jc013837, 2018.

Caniaux, G., Giordani, H., Redelsperger, J. L., Guichard, F., Key, E., and Wade, M.: Coupling between the Atlantic cold tongue and the West African monsoon in boreal spring and summer, J Geophys Res-Oceans, 116, C04003, https://doi.org/10.1029/2010jc006570, 2011.

Giordani, H. and Caniaux, G.: Diagnosing vertical motion in the Equatorial Atlantic, Ocean Dynam, 61, 1995-2018, https://doi.org/10.1007/s10236-011-0467-7, 2011.

Kopte, R., Brandt, P., Dengler, M., Tchipalanga, P. C. M., Macueria, M., and Ostrowski, M.: The Angola Current: Flow and hydrographic characteristics as observed at 11°S, J Geophys Res-Oceans, 122, 1177-1189, https://doi.org/10.1002/2016jc012374, 2017.

Körner, M., Brandt, P., and Dengler, M.: Seasonal cycle of sea surface temperature in the tropical Angolan upwelling system, EGUsphere, 1-33, https://doi.org/10.5194/egusphere-2022-973, 2022.

Nyadjro, E. S., Foli, B. A. K., Agyekum, K. A., Wiafe, G., and Tsei, S.: Seasonal Variability of Sea Surface Salinity in the NW Gulf of Guinea from SMAP Satellite, Remote Sens Earth Syst Sci, 5, 83-94, https://doi.org/10.1007/s41976-021-00061-2, 2022.

Radenac, M. H., Jouanno, J., Tchamabi, C. C., Awo, M., Bourles, B., Arnault, S., and Aumont, O.: Physical drivers of the nitrate seasonal variability in the Atlantic cold tongue, Biogeosciences, 17, 529-545, https://doi.org/10.5194/bg-17-529-2020, 2020.

Yang, Y., Wu, L. X., Cai, W. J., Jia, F., Ng, B., Wang, G. J., and Geng, T.: Suppressed Atlantic Nino/Nina variability under greenhouse warming, Nat Clim Change, 12, 814-821, https://doi.org/10.1038/s41558-022-01444-z, 2022.

Zeng, Z., Brandt, P., Lamb, K. G., Greatbatch, R. J., Dengler, M., Claus, M., and Chen, X.: Three-dimensional numerical simulations of internal tides in the Angolan upwelling region, J Geophys Res-Oceans, 126, e2020JC016460, https://doi.org/10.1029/2020JC016460, 2021.

---

## Author Comment (AC2)

Authors response to comments by Erik van Sebille to the manuscript "Physical processes in the upwelling regions of the tropical Atlantic" by Brandt et al. (pbrandt@geomar.de).
We would like to thank Erik van Sebille for the detailed and helpful comments to improve the manuscript. Below, we use black text for Erik van Sebille's comments and green text for our response.

RC2: 'Comment on egusphere-2022-1354', Erik van Sebille, 28 Feb 2023

Full disclosure: I am writing this review as Topic Editor for this manuscript. It has proven very difficult to find a second independent and knowledgeable reviewer for this manuscript, as most experts are already involved in the manuscript as authors. After discussions with two of the Chief Editors, I decided to write this second review as Topic Editor; focussing mostly on the effectiveness of the manuscript as a review article - Erik van Sebille

The topic of the manuscript (physical processes in the upwelling regions of the tropical Atlantic) surely warrants an in-depth review, and Ocean Science is an appropriate journal for it. The authors are from a wide and diverse range of institutes and expertises. This is all very strong and supports the trust in the review as unbiased and complete.

However, I have a few suggestions that I think will make the manuscript stronger and more impactful as a review article:

1. I encourage the authors to think about a schematic or figure that summarizes the key processes that are discussed in the paper. Such a figure might be hugely impactful, as it illustrates what the review is about

Difficult to find such a schematic. However, we improve and extended existing Figures to make the point of the review clearer.

2. The meridional extent of the region is never defined. While the zonal extent is discussed (line 54 and further), the meridional extend stays vague. Is it the 20S to 10N of figure 1?

We now removed the term "inner tropical Atlantic" and refer to the tropical Atlantic between 10°N and 20°S. The different upwelling systems are now marked in Fig.1.

3. Some of the modeling results (e.g. Figs 7, 8, 9) miss a clear reference. It is unclear now whether these figures have been created specifically for this manuscript, or come from another paper. In the first case, there should be much more discussion of the model setup etc (and it might be doubtful whether new/unpublished results fit in a review article like this); in the second case the captions need a clear reference.

We now state clearly the sources of the model output shown in Figs. 7, 8. and 9.

4. Section 5 misses figures. Whereas the other sections all have figures, this doesn't. Is that intentional? More generally, the style of the different sections varies quite a bit. I realize that each co-author was probably responsible for leading one section, but I would then still recommend carefully going through the text (and figures) to harmonize the style. That will help readers, and thus increase impact.

We included a new figure showing the interannual SST anomalies during the upwelling season of the three upwelling systems discussed (new Fig. 12). We also added the mean seasonal cycles of SST and Chlorophyll in the three upwelling systems in Fig. 1 and tried to harmonize the text as much as possible.

5. The manuscript tends to be fairly descriptive (answering what is happening) and relatively low on (physical) explanation. For example, line 601 states that "The zonal velocity field instead is dominated by the equatorial basin resonance of the 2nd and 4th baroclinic modes resulting in an EUC that vertically migrate largely independent of the thermocline (Brandt et al. 2016)" This sentence does not provide much information on _why_ it are the 2nd and 4th modes that are important here. That information may be in the reference, but the purpose of a review article is also to provide an accessible overview of the state-of-the-art; in this case (because of the title) also in terms of physical processes.

We are now more detailed regarding the explanation of the resonant equatorial basin modes of the 2nd and 4th baroclinic modes or the role of mixing in the TAUS for SST seasonality. Throughout the manuscript, we modified the text to better explain the different physical processes at work.

6. I somewhat missed a discussion of the similarities and differences with upwelling regions of the tropical Pacific and Indian Ocean. While the uniqueness of the basin is described well in the introduction; the extend to which the physical processes differ or agree in the other two basins is not very well discussed. I would expect that to be another purpose of such a review article?

In the final chapter, we added some open questions and future prospects regarding upwelling studies also relevant for the Pacific and Indian Ocean. We think that this will make the review article more attractive to a larger community.

Minor comments:

- line 14: 'inner' is a strange word here. 'Offshore'?

"inner" is removed throughout. We now refer to "the tropical Atlantic (10°N-20°S)"

- line 21: misses 'the' before 'northern boundary'

Changed

- line 22: remove comma after both

Changed

- line 22: nonlinearity in what?

It's the nonlinearity of the momentum equations, i.e., the inertia of the Guinea Current. We removed the term nonlinearity in the abstract and changed to "… role of the Guinea Current, its separation from the coast and the shape of the coastline …"

- Figure 1: I don't understand the units of mg m-3. Until what depth is this then? Would it not make more sense as mg m-2?

This is the unit provided by Copernicus-GlobColour. In the description at https://www.copernicus.eu/en/access-data/copernicus-services-catalogue/satellite-ocean-colour it is stated: Mass chlorophyll-a per unit of volume of near-surface water.

- line 82: the capitalization rules between tAUS and EAUS are not clear.

We changed tAUS to TAUS throughout.

- line 145: is the ecosystem indeed nitrate-limited? Would be good to provide a reference for that implicit assumption

The tropical Atlantic is mainly nitrate limited, see e.g., Moore et al. (2004), which now is included in the reference list. For that reason, model studies mostly focus on the nitrate budget and nitrate fluxes (see, e.g., Radenac et al. (2020)), and, when describing their results, we use nitrate as well. In general, we discuss nutrient supply in upwelling regions and changed therefore from nitrate supply to nutrient supply.

- Figure 3: why not use the same time period for all four datasets. Most of the period overlaps, but the start and end years are not consistent. It would be stronger to harmonize that

Now, all data used in Figs. 3 and 4 are averaged for the common period 1998-2020.

- line 194: is this the mean in time? Or the mean in space?

Changed to "The annual mean tropical cells in the central tropical Atlantic"

- line 203: 'helps to define' is vague wording here.

Changed to "is the strongest cooling term of the mixed layer heat budget during the onset of the ACT and sets"

- line 210: 'seasonally varying'

Changed

- line 231-234: so what is the conclusion then?

We included additional panels to Fig. 5 to show the seasonal variability of the EUC and changed the text as follows:

Radenac et al. (2020) indeed showed a different behaviour of the EUC during boreal spring and autumn, where a shallow EUC during boreal spring might prevent upward mixing of nitrate compared the deep phase of the EUC during boreal autumn, when nitrate more easily reach into the shear zone above the EUC core (Fig. 5). However, also the equatorial role being at maximum strength during boreal autumn (Heukamp et al., 2022) might contribute to the nitrate supply to the mixed layer by upwelling slightly south of the equator.

- line 268: why use the fourth baroclinic mode? What is so special about the fourth one?

We expanded the explanation of the resonant basin mode:
This behaviour can be associated with the resonance of the equatorial basin at the annual period. The period of a resonant equatorial basin mode is given by the total travel time of an equatorial Kelvin wave and its reflected equatorial Rossby wave. For the width of the equatorial Atlantic basin, the resonance period of the 4th baroclinic mode is close to the annual cycle. This basin mode is associated with maximum eastward velocity in the near-surface layer in boreal spring and maximum westward flow in boreal autumn (Brandt et al., 2016).

- line 309: which capes are meant with 'the capes'?

We clarified: Cape Three Points and Cape Palmas

- line 329-330: the 25C threshold for upwelling seems fairly arbitrary. The paper that it is based on in almost 50 years old. Has no newer research been done on this?

The threshold for upwelling is indeed arbitrary but is often used to define upwelling in the tropical Atlantic, see, e.g., Caniaux et al. (2011). However, we think that a reference for such an arbitrary threshold might not be appropriate and we removed the reference to Bakun (1978).

- line 487: what type of model was used by Zeng et al (2021)? A bit more description might help readers gauge the applicability of these conclusions

We now write: By applying a regional general circulation model forced solely by barotropic tides at the open boundaries, …

- line 513: this first sentence of section 5 is very vague. Which part of the upwelling? Variability of what?What is meant with 'It is the area and season most impacted [...]'?

We removed this sentence and started with a new figure showing the interannual SST anomalies during the upwelling season of the three upwelling systems discussed (new Fig. 12).

- line 519: 'boreal summer' is a biased and confusing term. While not simply write the names of the months?

We changed that to: It is most pronounced in the equatorial cold tongue east of 23°W and peaks during June-July (Keenlyside and Latif, 2007).

- line 543: what is meant with 'first' here?

We removed "first". It is not necessary.

- line 575: also provide a reference for the event in 2021?

We included a new Figure (Fig. 12) showing that event.

**References**

Bakun, A.: Guinea Current Upwelling, Nature, 271, 147-150, https://doi.org/10.1038/271147a0, 1978.

Brandt, P., Claus, M., Greatbatch, R. J., Kopte, R., Toole, J. M., Johns, W. E., and Böning, C. W.: Annual and semiannual cycle of equatorial Atlantic circulation associated with basin-mode resonance, J Phys Oceanogr, 46, 3011-3029, https://doi.org/10.1175/Jpo-D-15-0248.1, 2016.

Caniaux, G., Giordani, H., Redelsperger, J. L., Guichard, F., Key, E., and Wade, M.: Coupling between the Atlantic cold tongue and the West African monsoon in boreal spring and summer, J Geophys Res-Oceans, 116, C04003, https://doi.org/10.1029/2010jc006570, 2011.

Heukamp, F. O., Brandt, P., Dengler, M., Tuchen, F. P., McPhaden, M. J., and Moum, J. N.: Tropical instability waves and wind-forced cross-equatorial flow in the central Atlantic Ocean, Geophys Res Lett, 49, e2022GL099325, https://doi.org/10.1029/2022GL099325, 2022.

Keenlyside, N. S. and Latif, M.: Understanding equatorial Atlantic interannual variability, J Climate, 20, 131-142, https://doi.org/10.1175/Jcli3992.1, 2007.

Moore, J. K., Doney, S. C., and Lindsay, K.: Upper ocean ecosystem dynamics and iron cycling in a global three-dimensional model, Global Biogeochem Cy, 18, Gb4028, https://doi.org/10.1029/2004gb002220, 2004.

Radenac, M. H., Jouanno, J., Tchamabi, C. C., Awo, M., Bourles, B., Arnault, S., and Aumont, O.: Physical drivers of the nitrate seasonal variability in the Atlantic cold tongue, Biogeosciences, 17, 529-545, https://doi.org/10.5194/bg-17-529-2020, 2020.